# OOTSM: A Decoupled Linguistic Framework for Effective Scene Graph Anticipation

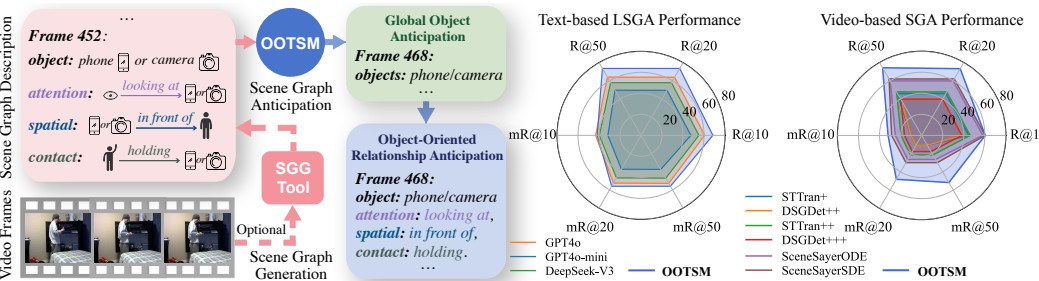

Figure 1: **Overview of OOTSM.** Our proposed **OOTSM** is a decoupled framework that effectively targets linguistic scene-graph anticipation, yet remains versatile through optional SGG integration for video inputs. The architecture comprises two principal components: **Global Object Anticipation (GOA)** predicts the future object set, and **Object-Oriented Relationship Anticipation (OORA)** forecasts object-centric relations. Radar plots summarize improvements on both tasks, especially at medium and long horizons (R@20, R@50).

## ABSTRACT

A scene graph is a structured representation of objects and their spatio-temporal relationships in dynamic scenes. Scene Graph Anticipation (SGA) involves predicting future scene graphs from video clips, enabling applications in intelligent surveillance and human-machine collaboration. While recent SGA approaches excel at leveraging visual evidence, long-horizon forecasting fundamentally depends on semantic priors and commonsense temporal regularities that are challenging to extract purely from visual features. We therefore propose **Linguistic Scene Graph Anticipation (LSGA)** as an independent forecasting task in the language domain, treating visual detection as a pluggable front-end while focusing on how temporal relational dynamics should be modeled and evaluated. Concretely, we introduce **Object-Oriented Two-Staged Method (OOTSM)**, an object-oriented two-stage framework that enhances LSGA through dynamic object prediction and relationship forecasting with temporal consistency constraints, accompanied by a dedicated LSGA benchmark derived from Action Genome annotations. Extensive experiments demonstrate that compact open-source Large Language Models (LLMs) fine-tuned for LSGA surpass strong zero-shot APIs (*i.e.*, GPT-4o, GPT-4o-mini, DeepSeek-V3) in most evaluation metrics under equivalent input conditions and context-window constraints. In particular, integration with frozen scene-graph detectors enables these LSGA advancements to yield superior video-based SGA performance, especially in long-horizon prediction scenarios (**+21.9**% mR@50), highlighting the substantial complementarity for the SGA task. Code is available at https://github.com/only4anonymous/OOTSM.

## 1 INTRODUCTION

Understanding relationships in human–scene interactions requires more than recognizing static information in a single frame; it necessitates comprehending complex interaction patterns that evolve among objects over time Mohamed et al. (2020); Wang et al. (2021). Recently, spatio-temporal scene graphs have emerged as a powerful structured representation of these dynamics by decomposing a scene into pairwise relations between humans and objects Xu et al. (2017); Ji et al. (2020); Li et al. (2022); Qiu et al. (2023). Because they already encode how interactions change over time, extending these graphs into the future is a valuable next step Liu et al. (2020); Peddi et al. (2024). This

predictive capability underpins diverse applications—for instance, anticipating abnormal behaviours in intelligent-surveillance footage Sarker et al. (2021); Chiranjeevi & Malathi (2024) or inferring user intent early in human–machine collaboration Zhang et al. (2024); Özdel et al. (2024).

To study this predictive problem more concretely, we therefore turn to focus on the recently proposed SGA challenge. Given a sequence of observed video frames, SGA aims to predict future scene graphs that capture both spatial and temporal relationships among objects. Despite its promise, existing SGA methods draw almost exclusively on visual cues contained in spatio-temporal scene graphs, limiting their ability to incorporate rich semantic priors for interaction prediction Yu et al. (2023); Li et al. (2025); Hsieh & Liu (2025), which can be vital for nuanced and long-horizon forecasting. This observation leads to a central question: how can we effectively capture and predict human-object interaction dynamics by combining visual evidence with commonsense priors?

Current approaches to SGA mostly utilize transformer-based architectures that jointly model temporal dependencies and spatial relationships from visual inputs Cong et al. (2021); Li et al. (2022); Mazzia et al. (2022); Feng et al. (2023). This joint optimization often entangles pixel level recognition with long-horizon relational reasoning, making it difficult to inject (or even measure) the role of semantic priors and commonsense regularities Chen et al. (2014); Vedantam et al. (2015); Wang et al. (2020); Heo & Kang (2023); Wu et al. (2023); Wang et al. (2025a;b); Zhao et al. (2025). Therefore, we argue in this paper that for SGA, an effective and modular approach is to separate visual processing and language processing into distinct components. By employing a frozen Scene Graph Generation (SGG) frontend for scene graph extraction and delegating temporal reasoning to a dedicated language model, each component can focus on its strengths—accurate visual representation and knowledge-informed prediction, respectively—while allowing for seamless integration of improved visual detectors without retraining the reasoning framework Chen et al. (2014); Gupta & Malik (2015); Vinyals et al. (2015); Antol et al. (2015).

Building on this decoupled strategy, we leverage an off-the-shelf tool for scene graph capturing (see *e.g.* Cong et al. (2021); Feng et al. (2023)) and concentrate on the language processing component, proposing a novel task called LSGA. Specifically, a LSGA task involves predicting future scene graphs given a sequence of scene graphs corresponding to a video clip (seen in fig. 1). By conducting its core anticipation reasoning primarily, LSGA uniquely positions LLMs to apply their extensive linguistic knowledge and learned world understanding to the SGA challenge. Our main contributions in this work are as follows:

- We formalize LSGA and release a dedicated benchmark by converting Action Genome into temporally ordered graph sequences with observed/predicted splits.

- We propose OOTSM—an effective framework that (i) predicts, for every future time-step, the set of objects that will appear or disappear, and (ii) refines each object's attention, spatial and contact relationships with a temporally consistent decoder.

- We describe a unified prompting and fine-tuning strategy that adapts compact open-source LLMs to the LSGA setting without requiring any visual feature, boosting a 3B-parameter Llama model from 49.5% to 73.6% R@20 on LSGA (**+24.1%**) while remaining within a single-GPU budget.

- When integrated with frozen scene-graph detectors, OOTSM improves video-based SGA performance, with long-term prediction showing mR@50 gains of **21.9%** and robustness evidenced by an average R@50 degradation of only **-0.68%** under 25% frame-error rates, validating the viability of linguistic temporal reasoning for anticipation scenarios.

## 2 RELATED WORK

**Scene Graph Generation.** SGG characterizes images through object-predicate graphs. Early approaches utilized frequency biases Zellers et al. (2018) or graph neural networks Xu et al. (2017) on Visual Genome Krishna et al. (2017), with later work addressing long-tail relations via contextual attention and debiasing strategies Tang et al. (2020). Our framework accepts SGG outputs as input, making detector improvements orthogonal to our method.

**Scene Graph Anticipation.** SGA extends SGG to temporal prediction over video streams. Current methods employ transformer pipelines with spatio-temporal attention Cong et al. (2021); Li et al. (2022) or model dynamics as continuous latent trajectories via SDEs Peddi et al. (2024). These

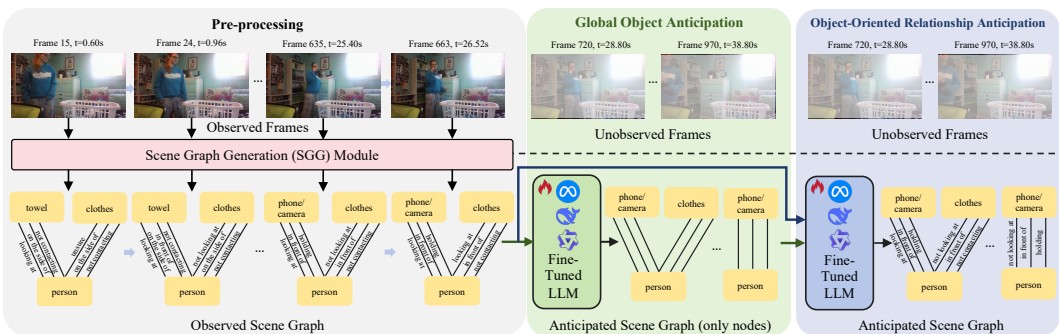

Figure 2: **Overall OOTSM pipeline.** The left branch converts observed video frames into compact textual scene graphs; **GOA** module performs dynamic object anticipation, whereas **OORA** module carries out object-oriented relationship anticipation. Additionally, scene graphs can bypass **GOA** for direct input into **OORA** (dotted arrow), provided the continuous-object constraint is maintained—only objects present in the final observed frame are projected forward. The dashed gray path is optional visual SGG tool integration.

approaches predominantly rely on visual cues and assume object persistence, limiting their ability to handle appearance/disappearance events.

**Large Language Models for Scene-Graph Tasks.** Recent work leverages LLMs to enhance scene graph semantics: SDSGG Chen et al. (2024) transfers linguistic priors for open-vocabulary predicates; VLPrompt Zhou et al. (2023) uses natural-language prompts for rare relations; and LLM4SGG Kim et al. (2024) parses descriptions into structured triples. For video, HyperGLM Nguyen et al. (2024) and Video-LLaMA Zhang et al. (2023) demonstrate multimodal reasoning capabilities without addressing future prediction. Our approach uniquely employs LLMs as the exclusive reasoning engine for long-term SGA.

## 3 METHOD

### 3.1 PRELIMINARY

Generally, the classical SGA setting takes a raw video prefix $F_{1:n}$ as input and requires a model to return a set of predicted scene graphs $\hat{G}_{n+1:T}$ that describe the objects and their pairwise relationships in the remaining frames. In practice, state-of-the-art pipelines realise this mapping with a single multi-modal network that must simultaneously decode visual patterns and reason over semantic structures—a design that conflates pixel-level recognition with symbolic anticipation.

We propose LSGA to effectively disentangle these two complementary aspects. Specifically, an external scene-graph detector first transforms the observed frames into a sequence of graphs $G_{1:n}$. The anticipation module then predicts $\hat{G}_{n+1:T}$ solely from this graph sequence, operating with its core anticipation reasoning entirely within the language domain and thereby leveraging the commonsense priors embedded in LLMs. Formally, while SGA learns a mapping $F_{1:n} \mapsto \hat{G}_{n+1:T}$, LSGA concentrates on the semantic sub-problem $G_{1:n} \mapsto \hat{G}_{n+1:T}$; consequently, the visual front-end can be substituted without necessitating retraining of the reasoning component.

Throughout the paper, we denote the unique set of objects (each coinciding with a single category) by $\mathcal{O} = \{o_1, \ldots, o_{N_O}\}$ and the set of relation types by $\mathcal{R} = \{r_1, \ldots, r_{N_R}\}$. Following Ji et al. (2020); Peddi et al. (2024), $\mathcal{R}$ is partitioned into three disjoint subsets: attention relations $\mathcal{R}_{\text{attn}}$, spatial relations $\mathcal{R}_{\text{spat}}$, and contact relations $\mathcal{R}_{\text{cont}}$. Each graph is therefore represented as a collection of triples (human, $o$, $r$) with $o \in \mathcal{O}$, $r \in \mathcal{R}$, providing a structured textual input that can be processed by any language model while remaining compatible with existing video annotations used later in the paper.

### 3.2 OVERALL FRAMEWORK

Our framework, OOTSM, introduces a two-stage framework for text-based LSGA that leverages LLM for future scene graph prediction (seen in fig. 2). To address context window constraints of

small-sized LLMs, which would otherwise lead to input truncation or necessitate larger models, we implement a two-stage framework. This framework first generates a global forecast of object occurrences (challenging due to object continuity limits, see Section A.3.1), followed by targeted object-oriented prompts for relation refinement—each remaining within token limitations. To optimize context utilization, we employ temporal graph merging, combining consecutive frames with identical relationships as follows:

$$G(F_t) \equiv G(F_{t+1}) \tag{1}$$

where $G(F_t)$ represents the scene graph of frame $t$. This aggregation focuses the model's attention on meaningful scene transitions rather than redundant patterns.

Expanding on this approach, we now detail the two-stage mechanism through which our anticipation framework operates: (i) First, **Global Object Anticipation (GOA)** module utilizes a fine-tuned LLM to generate holistic forecasts of object occurrences and their general relational patterns, though this global approach may not capture precise temporal transitions between individual relationships. (ii) Second, **Object-Oriented Relationship Anticipation (OORA)** module addresses context window constraints by generating targeted per-object prompts, enabling the model to refine each object's relational trajectory through multi-label classification with temporal continuity regularization, thus yielding more accurate relation predictions across the prediction horizon.

The final integration stage consolidates these object-level predictions into a coherent, temporally consistent scene graph, resolving potential conflicts and ensuring realistic interaction trajectories. For applications involving raw visual data, we provide an optional integration pathway leveraging existing SGG methodologies, such as STTran Cong et al. (2021), to bridge visual detections with our semantic anticipation capabilities, thereby enhancing applicability and interpretability across multimodal domains.

### 3.3 GLOBAL OBJECT ANTICIPATION

In the GOA module, we leverage the generative capabilities of a fine-tuned LLM to anticipate future scene graphs with a global perspective. Given observed video segments consolidated into coherent textual blocks $G_1, \ldots, G_n$, we construct a comprehensive prompt structured as:

$$P = H \oplus \text{ObservedText} \oplus \text{Instruction} \oplus \text{FutureFrameInfo} \tag{2}$$

where, $\oplus$ denotes string concatenation. $H$ provides task-oriented context, ObservedText encapsulates the temporal sequence of observed frames, Instruction delineates precise output formatting requirements, and FutureFrameInfo explicitly enumerates the future frames requiring predictions. A concrete example of GOA's prompt is provided in Section A.8.

Conventional uniform weighting treats all future frames equally, yet our experiments (table 3) show this approach compromises near-term accuracy as uncertain long-horizon predictions introduce high-variance gradients. We implement a cosine attenuation schedule for distant frames, reflecting their diminished

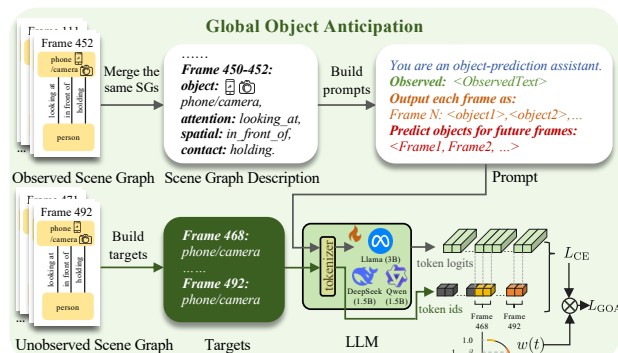

Figure 3: **GOA training flow.** Observed scene graphs with identical structures are first merged and converted to textual descriptions, then combined with instructions to construct prompts. A finetuned LLM subsequently predicts future object sets, supervised by unobserved scene graph targets via temporally weighted cross-entropy loss $L_{\text{GOA}}$.

signal-to-noise ratio and enabling the optimizer to prioritize higher-fidelity supervision. The loss for GOA is formulated as:

$$L_{\text{GOA}} = \frac{\sum_{t=n+1}^{T} w(t) \sum_{i=1}^{K_t} L_{\text{CE}}(p_{t,i}, y_{t,i})}{\sum_{t=n+1}^{T} w(t) K_t}, \tag{3}$$

where $t$ ranges over the future graphs $n+1, \ldots, T$, $w(t)$ is the temporal weight, and for each graph, $K_t$ denotes the number of byte-pair-encoded tokens in the target string representation of graph

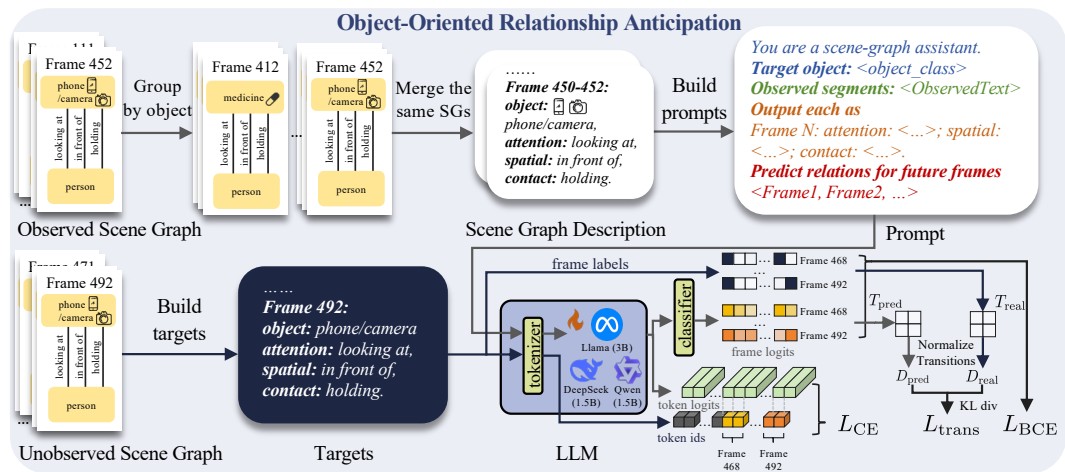

Figure 4: **OORA training flow.** Observed scene graphs are first grouped by object, then frames with identical scene graph structures are merged. Scene graph descriptions are combined with object-specific information to build prompts. These object-specific prompts guide a finetuned LLM to predict future relationships ($L_{\text{CE}}$). An auxiliary classifier generates per-frame relationship probabilities ($L_{\text{BCE}}$), while a transition regularizer ($L_{\text{trans}}$) ensures temporal coherence by penalizing improbable state transitions.

$G_t$, with predicted probabilities $p_{t,i}$ and ground-truth labels $y_{t,i}$ for $i = 1, \ldots, K_t$. The weighting function $w(t)$ is defined as:

$$w(t) = \beta \left[ 1 + \cos\left( \pi \frac{t - (n+1)}{T - (n+1)} \right) \right] + (1 - \beta), \tag{4}$$

where $\beta \in [0, 1]$ balances the base weight and cosine modulation.

GOA's predictions identify object occurrences and relational contexts temporally. Despite establishing global coherence, insufficient temporal granularity precludes precise scene graph construction, necessitating OORA module refinements.

### 3.4 OBJECT-ORIENTED RELATIONSHIP ANTICIPATION

OORA of our framework addresses a fundamental limitation of GOA: while GOA provides comprehensive scene-level forecasts, it lacks precision in tracking individual objects' temporal evolution. To resolve this, we implement object-oriented refinement, generating detailed multi-frame relationship distributions for each target entity. Beginning with the globally predicted scene graph $\widehat{G}^{(0)}$, we isolate the set of predicted future objects $\mathcal{O}$. For each object $o \in \mathcal{O}$, we craft an object-specific prompt:

$$P_o = H_o \oplus \text{ObservedSegments}_o \oplus \text{Instruction}_o \oplus \text{FutureFrames}_o \tag{5}$$

where $H_o$ provides the object-centric prediction context, ObservedSegments$_o$ succinctly summarizes the object's historical presence and interactions, Instruction$_o$ explicitly specifies the expected format of relationship predictions, and FutureFrames$_o$ enumerates the prediction horizon, enabling targeted temporal predictions. .

Relationship predictions in sequential frames often exhibit implausible transitions and temporal inconsistencies, severely compromising long-term forecasting reliability. To address this critical challenge, we introduce a temporal-transition regularizer $L_{\text{trans}}$ that penalizes improbable relationship state transitions across consecutive frames. Concretely, we first transform the frame-wise transition counts $T_{\text{real}}^r$ and the soft predictions $T_{\text{pred}}^r$ into normalized histograms $D_{\text{real}}^r$ and $D_{\text{pred}}^r$ (More details can be seen in Section A.1.1.), and then measure their discrepancy using the symmetrized Kullback–Leibler (KL) divergence:

$$L_{\text{trans}} = \frac{1}{N_R} \sum_{r=1}^{N_R} \frac{1}{2} \left( \text{KL} \left( D_{\text{pred}}^r \,\|\, D_{\text{real}}^r \right) + \text{KL} \left( D_{\text{real}}^r \,\|\, D_{\text{pred}}^r \right) \right), \tag{6}$$

Here, $T_{\text{real}}^r$ and $T_{\text{pred}}^r$ represent frame-wise relationship transition frequencies for ground-truth and predicted probabilities, respectively. This temporal-consistency prior discourages unnecessary high-frequency changes in relation trajectories and reduces error accumulation at long horizons.

To compute transition histograms, we employ an auxiliary discriminative classification layer trained with Binary Cross-Entropy loss $L_{\text{BCE}}$, which maps internal representations to relationship category probabilities.

Combining these discriminative losses with the generative teacher-forcing cross-entropy loss $L_{\text{CE}}$, we form the complete OORA training objective:

$$L_{\text{OORA}} = L_{\text{CE}} + L_{\text{BCE}} + \lambda L_{\text{trans}}, \tag{7}$$

where $L_{\text{CE}}$ denotes the teacher-forcing cross-entropy on the next-token distribution, $L_{\text{BCE}}$ supervises an auxiliary classification head that is trained independently of the frozen LLM parameters, and $L_{\text{trans}}$ enforces temporal smoothness of relationship transitions. A single hyper-parameter $\lambda$ controls the trade-off between temporal regularisation and the two accuracy-oriented terms. A single hyper-parameter is used to balance these terms; we set $\lambda = 0.03$ in experiments.

### 3.5 INTEGRATION FRAMEWORK FOR VIDEO SGA

We propose an optional integration pathway with established SGG tools to extend our textual SGA framework to raw video inputs, bridging visual perception and semantic anticipation. The process begins with frame-level detection, where SGG tools extract bounding boxes, object labels $\in \mathcal{O}$, and relationships $\in \mathcal{R}$, which are converted into our standardized textual format.

We implement temporal merging to consolidate consecutive frames with identical elements, optimizing for textual compactness without semantic redundancy. This condensed "observed text" serves as input to GOA's prompt for initial object anticipation, consistent with our primary pipeline. The anticipation then follows our two-stage methodology: the GOA module generates holistic future forecasts for the first stage while OORA provides object-centric refinements for the next stage.

Finally, we establish bidirectional mapping between textual predictions and spatial representations by linking semantic relationships to their corresponding bounding box indices. This produces a comprehensive spatiotemporal scene graph with both semantic and spatial localization data, enabling end-to-end video-based SGA. To validate robustness, we evaluate SGG noise impact using ground-truth scene graphs (table 1) and detector outputs (table 2). Our extensive robustness analysis (tables 5 to 8, seen in Section A.4) demonstrates OOTSM's resilience to SGG noise and detector quality variations. Additionally, to encourage high-confidence relation scores and suppress noise, we augment SGG training with a margin-based multilabel-threshold loss (detailed in Section A.1.2).

## 4 EXPERIMENT

**Dataset.** We follow the SGA evaluation protocol established by Peddi et al. (2024) on the Action Genome dataset Ji et al. (2020). Following standard preprocessing, we filter videos with fewer than three annotated frames, resulting in 11.4K videos total, and adopt the official train-test split. The dataset comprises 35 object classes and 25 relation classes categorized into attention, spatial, and contact relationships. We focus evaluation exclusively on Action Genome to provide controlled assessment of temporal reasoning capabilities and enable direct comparison with existing SGA methods.

**Evaluation Metrics.** Performance evaluation for all experiments employs the Recall@K (R@K) and meanRecall@K (mR@K) metrics, where K is chosen from 10, 20, 50. Recall@K measures the model's ability to anticipate relevant relationships between observed objects in future frames, while meanRecall@K addresses the issue of relationship class imbalance by equally weighing the predictive performance across all predicate classes. To provide a comprehensive evaluation, we vary the fraction of initial observed context frames, denoted as $\mathcal{F} \in 0.3, 0.5, 0.7, 0.9$, examining both short-term and long-term anticipation capabilities.

**Text-based LSGA Setting.** For the pure textual SGA scenario, predictions of future frames are made directly using ground-truth annotations from observed frames. Key LLM fine-tuning parameters are detailed in Section A.2. We evaluate our framework, OOTSM, by comparing its performance against two groups of models:

- **Direct API-based Methods:** Models using direct API inference without task-specific fine-tuning, including GPT-4o, GPT-4o-mini, and DeepSeek-3V.

- **Alternative Backbone LLMs within Our Framework:** Different backbone language models integrated within our framework, OOTSM, specifically Llama-3.2-3B Instruct, DeepSeek-R1-1.5B, and Qwen 2.5-1.5B Instruct.[1]

**Video-based SGA Setting.** We also assess our model in the Grounded Action Genome Scenes (GAGS) scenario, which provides the most comprehensive scene information through precise bounding boxes and object category labels. This setting minimizes challenges from occlusions and image quality issues, thus directly evaluating our model's reasoning capability more effectively. In this setting, we benchmark our framework against state-of-the-art baseline methods, including STTran+ Cong et al. (2021), DSGDet+ Feng et al. (2023), STTran++ Cong et al. (2021), DSGDet++ Feng et al. (2023), SceneSayerODE Peddi et al. (2024), and SceneSayerSDE Peddi et al. (2024).

Table 1: **Results for text-based LSGA (GT scene graphs, without SGG tool)**. Evaluation is based on Recall@K and meanRecall@K with $K = 10, 20, 50$. We compare two settings: (1) **w/o GOA**: relation prediction conditioned on observed GT objects; (2) **w/ GOA**: future object prediction followed by per-object relation reasoning. **Bold** indicates the best result, and underline indicates the second-best result.

| Setting | Method | R@10 | R@20 | R@50 | mR@10 | mR@20 | mR@50 |
|---------|--------|------|------|------|-------|-------|-------|
| w/o GOA | GPT4o-mini | 47.2 | 49.2 | 49.2 | 32.6 | 38.2 | 38.3 |
|  | GPT4o | 61.2 | 64.6 | 64.7 | **43.7** | 52.9 | 53.2 |
|  | DeepSeek-V3 | 55.2 | 57.9 | 58.0 | 38.9 | 46.4 | 46.6 |
|  | OOTSM (DeepSeek-R1-1.5B) | 60.7 | 64.4 | 64.4 | 35.8 | 43.5 | 43.5 |
|  | OOTSM (Qwen2.5-1.5B Instruct) | 62.0 | 65.1 | 65.1 | 36.2 | 43.7 | 43.7 |
|  | OOTSM (Llama-3.2-3B Instruct) | **68.7** | 72.4 | 72.4 | 42.8 | 51.8 | 51.8 |
| w/ GOA | GPT4o-mini | 47.5 | 49.5 | 49.6 | 31.6 | 37.3 | 37.5 |
|  | GPT4o | 60.6 | 63.9 | 64.0 | 43.3 | 52.5 | 52.8 |
|  | DeepSeek-V3 | 55.1 | 57.9 | 57.9 | 39.7 | 46.8 | 47.1 |
|  | OOTSM (DeepSeek-R1-1.5B) | 61.0 | 65.6 | 65.6 | 35.3 | 44.2 | 44.2 |
|  | OOTSM (Qwen2.5-1.5B Instruct) | 61.8 | 65.3 | 65.3 | 36.3 | 43.8 | 43.8 |
|  | OOTSM (Llama-3.2-3B Instruct) | 68.4 | **73.6** | **73.6** | 41.5 | **56.0** | **56.0** |

## 4.1 TEXT-BASED LSGA RESULTS

Table 1 reports Recall@K and meanRecall@K for text-based LSGA under two settings—w/o GOA (direct relation prediction, assuming all objects in the last observed frame persist) and w/ GOA (object forecasting before relation reasoning). In both cases, recall jumps markedly from R@10 to R@20, reflecting performance across short-, mid-, and long-term horizons.

Our proposed OOTSM (Llama-3.2-3B Instruct) achieves the highest R@20/50: 72.4%/72.4% without GOA and 73.6%/73.6% with GOA. GPT-4o is competitive, recording the highest mR@10 (43.7%) without GOA. Adding predicted objects with GOA increases OOTSM's mR@20 by 4.2% over the single-stage setting, indicating that GOA refines the candidate object set and improves long tail prediction consistency. Further analyses are provided in Sections A.4 to A.7 and A.8.3.

## 4.2 VIDEO-BASED SGA RESULTS

Results for video-based SGA under constrained conditions (table 2) demonstrate our OOTSM method's consistent superiority across all observation fractions($\mathcal{F} = 0.3, 0.5, 0.7, 0.9$). When compared to strong baselines (*e.g.*, SceneSayerSDE), our approach exhibits substantial performance advantages, particularly in medium-to-long-range predictions at R@20 and R@50. At $\mathcal{F} = 0.9$, OOTSM achieves significant improvements of 11.3% and 12.3% in R@20 and R@50, respectively, relative to SceneSayerSDE. The pronounced enhancements in mR metrics further demonstrate our method's capacity for effective long-tail relation prediction through language prior integration.

While short-term predictions (R@10) show modest improvements, this reflects language constraints' primary benefit for extended horizons. Robustness analysis (tables 5 to 8, detailed in Section A.4) shows minimal R@50 degradation (average -0.68%) even under 25% frame error rates, highlighting OOTSM's superior long-term resilience. Additionally, temporal reasoning validation (Section A.3.2)

---

[1]Due to compute limits, we fine-tune $\leq$3B backbones on a single GPU; larger LLMs are API-only baselines.

Table 2: **Results for SGA of GAGS under the with constraint setting (utilizing SGG tool).** Evaluation is conducted with different observed context fractions ($\mathcal{F}$). We report Recall@10/20/50 and meanRecall@10/20/50.

| $\mathcal{F}$ | Method | R@10 | R@20 | R@50 | mR@10 | mR@20 | mR@50 |
|---|---|---|---|---|---|---|---|
| 0.3 | STTran+ | 30.8 | 32.8 | 32.8 | 7.1 | 7.8 | 7.8 |
| | DSGDet+ | 27.0 | 28.9 | 28.9 | 6.7 | 7.4 | 7.4 |
| | STTran++ | 30.7 | 33.1 | 33.1 | 11.8 | 13.3 | 13.3 |
| | DSGDet++ | 25.7 | 28.2 | 28.2 | 11.1 | 12.8 | 12.8 |
| | SceneSayerODE | 34.9 | 37.3 | 37.3 | 15.1 | 16.6 | 16.6 |
| | SceneSayerSDE | **39.7** | 42.2 | 42.3 | 18.4 | 20.5 | 20.5 |
| | **OOTSM w/o GOA (Ours)** | 38.8 | **48.3** | **49.3** | 19.5 | **31.5** | **33.7** |
| | **OOTSM w GOA (Ours)** | 39.0 | 47.8 | 48.4 | **19.7** | **31.5** | 32.3 |
| 0.5 | STTran+ | 35.0 | 37.1 | 37.1 | 8.0 | 8.7 | 8.8 |
| | DSGDet+ | 31.2 | 33.3 | 33.3 | 7.8 | 8.6 | 8.6 |
| | STTran++ | 35.6 | 38.1 | 38.1 | 15.2 | 17.8 | 15.2 |
| | DSGDet++ | 29.3 | 31.9 | 32.0 | 13.9 | 20.6 | 13.9 |
| | SceneSayerODE | 40.7 | 43.4 | 43.4 | 17.4 | 19.2 | 19.3 |
| | SceneSayerSDE | **45.0** | 47.7 | 47.7 | 20.7 | 23.0 | 23.1 |
| | **OOTSM w/o GOA (Ours)** | 44.9 | 50.1 | 50.1 | **23.1** | 29.9 | 29.9 |
| | **OOTSM w GOA (Ours)** | 43.4 | **52.3** | **52.8** | 21.6 | **33.8** | **35.9** |
| 0.7 | STTran+ | 40.0 | 41.8 | 41.8 | 9.1 | 9.8 | 9.8 |
| | DSGDet+ | 35.5 | 37.3 | 37.3 | 9.6 | 9.6 | 9.6 |
| | STTran++ | 41.3 | 43.6 | 43.6 | 18.2 | 18.2 | 18.2 |
| | DSGDet++ | 33.9 | 36.3 | 36.3 | 15.9 | 15.9 | 15.9 |
| | SceneSayerODE | 49.1 | 51.6 | 51.6 | 21.0 | 22.9 | 22.9 |
| | SceneSayerSDE | 52.0 | 54.5 | 54.5 | 24.1 | 26.5 | 26.5 |
| | **OOTSM w/o GOA (Ours)** | 52.6 | 56.9 | 56.9 | 26.6 | 32.9 | 32.9 |
| | **OOTSM w GOA (Ours)** | **53.6** | **58.9** | **58.9** | **26.8** | **34.8** | **34.9** |
| 0.9 | STTran+ | 44.7 | 45.9 | 45.9 | 10.3 | 10.8 | 10.8 |
| | DSGDet+ | 38.8 | 40.0 | 40.0 | 10.2 | 10.7 | 10.7 |
| | STTran++ | 46.0 | 47.7 | 47.7 | 19.6 | 21.4 | 21.4 |
| | DSGDet++ | 38.1 | 39.8 | 39.8 | 16.3 | 17.7 | 17.7 |
| | SceneSayerODE | 58.1 | 59.8 | 59.8 | 25.0 | 26.4 | 26.4 |
| | SceneSayerSDE | 60.3 | 61.9 | 61.9 | 28.5 | 29.8 | 29.8 |
| | **OOTSM w/o GOA (Ours)** | **61.2** | 69.8 | 70.1 | 31.3 | 43.2 | 43.8 |
| | **OOTSM w GOA (Ours)** | 60.6 | **73.2** | **74.2** | **31.9** | **48.1** | **51.7** |

confirms OOTSM learns genuine temporal dynamics rather than exploiting superficial patterns, validating the method's effectiveness.

Table 3: **Ablation on weighting and transition regularizers. Weight** refers to the *cosine-weighted CE loss* applied in GOA. **Transition** denotes the *transition loss* introduced in OORA.

| Index | Weighting | Transition | R@10 | R@20 | R@50 | mR@10 | mR@20 | mR@50 |
|---|---|---|---|---|---|---|---|---|
| (a) | — | — | 67.5 | 72.6 | 72.6 | 41.3 | 53.9 | 53.9 |
| (b) | — | ✓ | 68.1 | 73.3 | 73.3 | 41.4 | 54.9 | 54.9 |
| (c) | ✓ | — | 67.9 | 73.3 | 73.3 | **42.0** | 55.6 | 55.6 |
| (d) | ✓ | ✓ | **68.4** | **73.6** | **73.6** | 41.5 | **56.0** | **56.0** |

### 4.3 IMPACT OF COSINE–WEIGHTED AND TRANSITION LOSSES

We evaluate two temporal regularizers on the validation split. Cosine-weighted CE loss is applied in GOA; each token loss is rescaled by the decay factor of eq. (4) with $\beta = 0.5$. Transition loss is the gated KL term introduced in OORA and weighted by $\lambda = 0.03$. The BCE used to train the classifier head in OORA does not update the LLM backbone and is therefore omitted from this ablation.

The results are shown as table 3. Adding the cosine weighting lifts overall recall from 72.6% to 73.3% and increases the long-tail metric mR@20 by 1.7% (c). The gains confirm that down-weighting distant-future tokens reduces noisy gradients and lets the model emphasize near-term, high-confidence patterns, markedly improving tail recall. Using the transition loss alone also reaches 73.3% R@20,

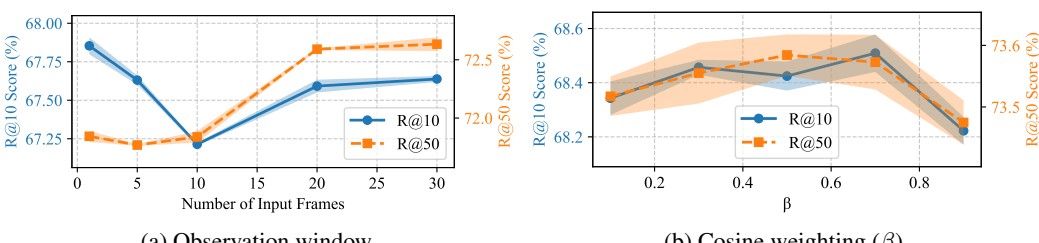

(a) Observation window.        (b) Cosine weighting ($\beta$).

Figure 5: **Observation window length and weighting hyper-parameter.** Left: effect of varying the number of observed frames on recall performance. Right: study of the cosine-weighting coefficient $\beta$. Shaded region denotes one standard deviation over three independent runs per value.

but improves mR@20 only to 54.9% (b), suggesting it mainly enforces smoothness on frequent relations. Combining both regularizers (d) further raises overall recall to 73.6% and yields the best balanced score of 56.0% mR@20, showing that the two losses are complementary: cosine weighting broadens tail coverage, while the transition loss stabilises common relational patterns. The slight drop in mR@10 compared with row c indicates that excessive smoothing can still dampen legitimate abrupt changes in very short windows, motivating future adaptive gating strategies.

### 4.4 Effect of the Observation Window

Figure 5a plots short-term (R@10) and long-term (R@50) prediction accuracy as the number of input observation frames increases from 1 to 30. When only a few frames are observed, the model achieves its highest R@10 but exhibits relatively poor R@50. As more frames are added, R@50 recovers significantly and then plateaus, while R@10, after a small initial dip, gradually returns to its original level. In summary, adding history frames yields clear gains in long-range prediction but diminishing returns for short-range accuracy. Nevertheless, to maximize available temporal context, we feed as many frames as possible—subject to a 2k token limit—to strike a balance between predictive performance and inference latency.

### 4.5 Effect of the weighting hyper-parameter

As shown in fig. 5b, we examine GOA's training sensitivity to the cosine-weighting coefficient $\beta$, which modulates future token loss down-weighting. The performance curve exhibits a shallow U-shape across prediction horizons. Very small $\beta$ values concentrate loss on near-term frames, causing overfitting to short-range patterns and degrading overall performance. Moderate $\beta$ values balance high-confidence short-range supervision with sufficient long-range exposure to regularize temporal dynamics, yielding monotonically improving recall. Beyond optimal levels, near-uniform weighting reintroduces noisy gradients from uncertain far-future tokens, reverting metrics toward their low-$\beta$ baseline. Our findings indicate that moderate decay ($0.3 < \beta < 0.7$) optimally balances near-future reliability with comprehensive horizon coverage.

## 5 Conclusion

This work introduces OOTSM, a novel two-stage, language-driven framework that fundamentally reframes scene-graph anticipation as textual inference. Our fine-tuned LLM with cosine-weighted and transition loss demonstrates state-of-the-art performance by consistently outperforming visual baselines on both our pioneering text-only LSGA benchmark and standard SGA evaluation, particularly at extended prediction horizons. These advantages stem from the principled separation of object and relation reasoning within a linguistic interface that naturally produces unordered multi-label triples, accommodates object dynamics, and effectively leverages linguistic associations when visual evidence is ambiguous or incomplete. Future directions include developing end-to-end video integration that eliminates modality transitions, implementing open-vocabulary object representation to address long-tail recognition failures and enhance generalization across varied scenarios, and exploring adaptive temporal constraints to balance between stability and legitimate state changes.

## ETHICS STATEMENT

We study LSGA for video-based SGA using only publicly available datasets under their licenses. We collect no new data, involve no human subjects, and do not process personally identifiable or sensitive attributes; annotations are at the object–relation level (*e.g.*, person–object interactions), not identities. Scene-graph methods are dual-use: potential risks include privacy concerns in public video and biases inherited from datasets or detectors. We document intended use and limitations in the repository, discourage identity/protected-attribute inference, and recommend legal/consent compliance and basic bias/shift checks before deployment. The authors have no conflicts of interest.

## REPRODUCIBILITY STATEMENT

We provide all assets needed to reproduce our results at the anonymous repository `https://github.com/only4anonymous/OOTSM`: data-to-graph conversion scripts (Action Genome → temporally ordered SG sequences), split generation, prompt templates, training and inference scripts, and configuration files with all hyperparameters. Section 4 and Appendix A.2 detail model variants and procedures; Appendix A.8 and Appendix A.1 describe preprocessing and implementation details. All baselines use public implementations or documented APIs; for API baselines we include scripts to reproduce queries or evaluate cached predictions (noting rate-limit/cost constraints). Hardware and efficiency measurements are also reported. These materials enable independent reproduction of tables 1 and 2 and our ablations.

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

# A  APPENDIX

APPENDIX CONTENTS

## A.1 Implementation and Mathematical Details

This section provides detailed mathematical derivations for two key technical components mentioned in the main text: the temporal transition regularizer and the enhanced multi-label supervision mechanism. The complete mathematical frameworks and implementation details are presented in Sections A.1.1 and A.1.2 respectively to enable reproducible results.

### A.1.1 Detailed Derivation of the Transition Regularizer

This section provides a detailed derivation of the temporal transition regularizer ($L_{\text{trans}}$), introduced in OORA to encourage realistic temporal dynamics in relationship predictions. For each relationship category $r \in [1, N_R]$, we first compute two transition matrices, capturing transitions between consecutive frames. Specifically, the ground-truth transition count matrix $T_{\text{real}}^r \in \mathbb{R}^{2 \times 2}$ is defined as:

$$T_{\text{real}}^r(i, j) = |\{t : y_r(t) = i, y_r(t+1) = j\}|, \tag{8}$$

where $y_r(t) \in \{0, 1\}$ indicates the presence (1) or absence (0) of relationship $r$ at graph $g_t$. Correspondingly, the predicted transition matrix $T_{\text{pred}}^r \in \mathbb{R}^{2 \times 2}$ accumulates predicted probabilities:

$$T_{\text{pred}}^r(i, j) = \sum_t p_r^{(i)}(t) \, p_r^{(j)}(t+1), \tag{9}$$

with $p_r^{(i)}$ being the predicted probability of relationship $r$ at state $i$ at time step $\tau$.

These matrices are normalized to produce valid probability distributions:

$$D_{\text{real}}^r = \frac{T_{\text{real}}^r}{|T_{\text{real}}^r|_1 + \varepsilon}, \quad D_{\text{pred}}^r = \frac{T_{\text{pred}}^r}{|T_{\text{pred}}^r|_1 + \varepsilon}, \tag{10}$$

where a small term $\varepsilon = 10^{-9}$ is added to maintain numerical stability.

To avoid instability arising from sparse relationships and minor fluctuations in predicted probabilities, we introduce two gating thresholds: a *count threshold* $\delta$ and a *probability-change threshold* $\tau$. Specifically, the threshold $\delta$ filters out relationships lacking sufficient observed transitions:

$$|T_{\text{real}}^r|_1 > \delta, \tag{11}$$

whereas $\tau$ ensures only substantive changes between consecutive predictions contribute to the predicted transition matrix:

$$|p_r(t+1) - p_r(t)| > \tau. \tag{12}$$

Empirically, we select $\tau = 0.2$ to exclude trivial fluctuations in predicted probabilities, thereby enhancing robustness.

With these thresholds, we define the temporal transition loss formally using the symmetrized Kullback–Leibler (KL) divergence, averaged across valid relationship categories:

$$L_{\text{trans}} = \frac{1}{|\mathcal{R}_\delta|} \sum r \in \mathcal{R}_\delta \frac{\text{KL}(D_{\text{pred}}^r | D_{\text{real}}^r) + \text{KL}(D_{\text{real}}^r | D_{\text{pred}}^r)}{2}, \tag{13}$$

where the set of valid relationships is given by $\mathcal{R}_\delta = r : |T_{\text{real}}^r|_1 > \delta$.

For efficient implementation, we vectorize this computation across batches and relationships, applying the aforementioned masking based on $\delta$ and $\tau$. The KL divergence between two distributions $P$ and $Q$ is computed explicitly as:

$$\text{KL}(P | Q) = \sum_{i \in 0, 1} P_i \log \frac{P_i}{Q_i}. \tag{14}$$

In practice, we set the regularization strength hyperparameter $\lambda = 0.03$ to achieve a balanced trade-off between promoting smooth, plausible transitions and allowing the model sufficient flexibility for dynamic interactions.

Moreover, to enable the computation of the temporal-transition regularizer $L_{\text{trans}}$, we introduce an auxiliary discriminative classification layer atop the frozen LLM backbone, mapping internal object-frame representations to logits $\mathbf{z} \in \mathbb{R}^{N_R}$ per relationship category. These logits are subsequently converted to probabilities via a sigmoid function:

$$\mathbf{p} = \sigma(\mathbf{z}), \tag{15}$$

where each element $p_r$ denotes the probability that relationship $r$ occurs in a given frame. The discriminative head parameters are optimized separately from the frozen backbone using a Binary Cross-Entropy (BCE) loss:

$$L_{\text{BCE}} = -\frac{1}{N_R} \sum_{r=1}^{N_R} \left[ y_r \log p_r + (1 - y_r) \log(1 - p_r) \right]. \tag{16}$$

This auxiliary discriminative supervision facilitates stable estimation of transition frequencies required for calculating $D_{\text{pred}}^r$ and $D_{\text{real}}^r$—the normalized histograms employed by our transition regularizer.

### A.1.2 Enhanced Multi-label Supervision for the SGG Tool

In preliminary experiments we observed that the off-the-shelf SGG detector tends to assign ambiguous confidence values clustered around 0.5, especially for spatial and contacting relations that often co-occur. Such blurred predictions propagate to GOA and OORA modules, degrading both object filtering and downstream temporal reasoning. To sharpen the detector's decision boundary we re-trained its relation head with a *threshold-aware margin loss* that complements the conventional BCE.

Let $\mathbf{p} \in [0,1]^{N_R}$ be the sigmoid outputs for a given object pair and let $\mathbf{y} \in \{0,1\}^{N_R}$ be the corresponding multi-hot ground-truth vector. We impose a positive margin $\gamma_{\text{pos}}$ for present relations and a negative margin $\gamma_{\text{neg}}$ for absent ones:

$$L_{\text{thr}}(\mathbf{p}, \mathbf{y}) = \frac{1}{N_R} \sum_{r=1}^{N_R} \Big[ y_r \, \max\big(0, \gamma_{\text{pos}} - p_r\big) + \big(1 - y_r\big) \max\big(0, p_r - \gamma_{\text{neg}}\big) \Big], \tag{17}$$

with $\gamma_{\text{pos}} = 0.9$ and $\gamma_{\text{neg}} = 0.5$. The hinge-style clamps encourage confident positives ($p_r > 0.9$) while pushing negatives below 0.5, thereby carving out an explicit decision margin in the interval $[0.5, 0.9]$. The final training objective for each relation category becomes

$$L_{\text{rel}} = L_{\text{BCE}} + \eta \, L_{\text{thr}}, \tag{18}$$

where $\eta = 0.5$ balances likelihood fitting and margin enforcement. During label preparation we extend the one-hot encoder to accommodate genuine multi-label annotations: if the dataset provides a list of relation indices for an object pair, we set all corresponding positions in $\mathbf{y}$ to 1; otherwise we keep the standard single-label assignment. This homogeneous encoding allows the detector to learn single and compound relations within a unified framework. At inference time, we apply a probability threshold of 0.6 on the sigmoid outputs, such that any relation dimension with $p_r > 0.6$ is considered as a positive prediction.

### A.2 Detailed Experimental Settings

All experiments described in this paper were conducted under the following comprehensive configuration to ensure reproducibility and systematic evaluation:

**Hardware and Environment.** We trained and evaluated all models using a single NVIDIA A100 GPU with 40GB memory, providing sufficient computational resources for handling the extensive temporal context required in long-horizon scene graph anticipation tasks.

**Training Configuration.** For both GOA and OORA, we employed SGD optimizer with a fixed learning rate of $1 \times 10^{-5}$. The batch size was set to 1, optimizing each example individually due to memory constraints imposed by the substantial input sequence lengths. For all experiments involving LLM fine-tuning, we applied Low-Rank Adaptation (LoRA) with $rank = 32$ and $\alpha = 32$, training GOA for 5 epochs and OORA for 10 epochs to achieve optimal convergence across both stages.

**Input and Output Specifications.** To maximize temporal context for the LLM-based anticipation tasks, we employed a context window length of 2048 tokens maximum. The input formatting follows structured prompt templates as detailed in Appendix A.9. For output generation, we implemented observation-ratio-dependent length constraints to balance prediction comprehensiveness with computational efficiency: in the 30%, 50%, 70%, and 90% observation settings, we set maximum output lengths of 1280, 1024, 768, and 512 tokens respectively for GOA, and 1792, 1536, 1280, and 1024 tokens respectively for OORA. These asymmetric constraints reflect the inherently more complex relational structure predictions required in the OORA stage.

**Decoding Strategy.** During inference, we applied nucleus sampling with temperature=0.7 and top-p=0.4 to balance generation diversity with coherence, particularly important for analytical evaluation where both creativity and consistency in predictions are valued.

These settings are consistently applied across all reported experiments, including evaluations with different backbone models and ablation studies, ensuring uniform comparison conditions and facilitating reproducibility of our results.

### A.3 Motivations for Two-Stage Framework Design

This section establishes the theoretical foundations and empirical justifications for our proposed two-stage OOTSM framework through comprehensive experimental analyses. We demonstrate the design necessity through examining the limitations of traditional continuous-object assumptions in Section A.3.1, followed by diagnostic validation of our framework's temporal reasoning capabilities in Section A.3.2.

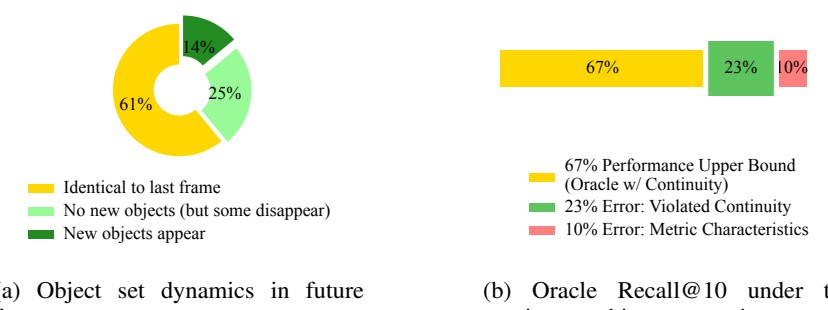

(a) Object set dynamics in future frames.

(b) Oracle Recall@10 under the continuous-object assumption.

Figure 6: **Analysis of the continuous-object assumption.** (a): Statistical distribution of object set changes between the last observed frame and future prediction frames (90% observation, 10% prediction). (b): Impact on an oracle Recall@10 experiment, demonstrating performance limitations imposed by strictly adhering to object continuity.

### A.3.1 Limitations of the Continuous-Object Assumption

The conventional continuous-object assumption in scene graph anticipation posits that objects present in the final observed frame persist throughout future frames. Our analysis reveals significant limitations of this assumption, as illustrated in fig. 6.

Our statistical analysis demonstrates that the continuous-object assumption fails to capture the dynamic nature of real-world scenes. When evaluating under the standard 90% observation, 10% prediction protocol on the Action Genome dataset, we observe that only 61% of video sequences maintain object consistency between the final observed frame and subsequent prediction frames. The remaining 39% exhibit significant object dynamics: 14% introduce previously unseen objects, while 25% show object disappearances from the scene.

To quantify the performance impact of this assumption, we conducted an oracle experiment measuring the maximum achievable Recall@10 when constrained by the continuous-object assumption. Even with perfect relationship prediction for the objects present in the final observed frame, the ceiling performance reaches only 67%. This represents a total performance gap of 33% compared to the theoretical optimal performance, with 23% attributable to violated object continuity assumptions and the remaining 10% due to evaluation metric constraints (the Recall@K metric limits numerator predictions to K relationships while normalizing by the total number of ground truth relationships).

These findings underscore the critical importance of modeling dynamic object appearances and disappearances in scene graph anticipation tasks, providing strong motivation for our proposed object anticipation approach in OOTSM.

### A.3.2 Validation of Temporal Reasoning Effectiveness

To verify that our proposed OOTSM framework genuinely learns temporal dynamics rather than achieving superficial gains through simple final-frame replication, we conducted comprehensive evaluations on the SGA task, comparing OOTSM against naive strategies and strong visual baselines across observation ratios $\mathcal{F} \in 0.3, 0.5, 0.7, 0.9$. The experimental results are presented in table 4.

As the results demonstrate, OOTSM achieves larger improvements in mR metrics compared to standard R metrics, indicating better performance on long-tail categories and newly appearing objects. Standard recall metrics show improvements of 6.5% (R@20) and 7.0% (R@50), while class-balanced mean recall metrics demonstrate gains of 12.1% (mR@20) and 13.7% (mR@50). This disparity becomes more apparent under the $\mathcal{F} = 0.9$ configuration, where mR improvements (18.3% and 21.9%) exceed standard recall improvements (11.3% and 12.3%). The macro-averaged mR metrics equalize the contribution of rare relationship categories regardless of frequency, suggesting that our approach leverages linguistic priors to predict infrequent relationships that visual methods typically miss.

The method shows greater improvements at larger K values, indicating stronger performance in medium to long-term predictions. While R@10 improvements remain modest across different observation ratios, R@20 and R@50 demonstrate consistent gains. Under the $\mathcal{F} = 0.9$ configuration, R@20 and R@50 improvements (11.3% and 12.3%) exceed typical short-term performance gains. This pattern suggests that linguistic reasoning becomes more effective for longer prediction horizons, likely due to two factors: (i) LLMs can leverage learned patterns about temporal object dynamics, and (ii) scene graphs provide structured representations with reduced noise compared to raw visual features, offering better signal for long-term inference.

Table 4: **Validation of temporal reasoning effectiveness across observation ratios.** † denotes the naive strategy (predicting all future frames using the last observed frame's scene graph).

| $\mathcal{F}$ | Method | R@10 | R@20 | R@50 | mR@10 | mR@20 | mR@50 |
|---|---|---|---|---|---|---|---|
| 0.3 | SceneSayerSDE† | 38.7 | 42.0 | 42.1 | 16.6 | 19.9 | 19.9 |
|  | SceneSayerSDE | 39.7 | 42.2 | 42.3 | 18.4 | 20.5 | 20.5 |
|  | OOTSM | 39.0 | 47.8 | 48.4 | 19.7 | 31.5 | 32.3 |
| 0.5 | SceneSayerSDE† | 43.4 | 47.1 | 47.1 | 18.5 | 21.9 | 22.1 |
|  | SceneSayerSDE | 45.0 | 47.7 | 47.7 | 20.7 | 23.0 | 23.1 |
|  | OOTSM | 43.4 | 52.3 | 52.8 | 21.6 | 33.8 | 35.9 |
| 0.7 | SceneSayerSDE† | 50.7 | 53.9 | 53.9 | 21.9 | 25.2 | 25.2 |
|  | SceneSayerSDE | 52.0 | 54.5 | 54.5 | 24.1 | 26.5 | 26.5 |
|  | OOTSM | 53.6 | 58.9 | 58.9 | 26.8 | 34.8 | 34.9 |
| 0.9 | SceneSayerSDE† | 59.3 | 61.5 | 61.5 | 26.5 | 29.2 | 29.2 |
|  | SceneSayerSDE | 60.3 | 61.9 | 61.9 | 28.5 | 29.8 | 29.8 |
|  | OOTSM | 60.6 | 73.2 | 74.2 | 31.9 | 48.1 | 51.7 |
| Average $\Delta$ (vs SceneSayerSDE†) | SceneSayerSDE | +1.3 | +0.5 | +0.4 | +2.0 | +0.9 | +0.9 |
|  | OOTSM | +1.2 | +7.0 | +7.4 | +4.1 | +13.0 | +14.6 |

Table 5: **Effect of noise type on short-term OOTSM performance (R@10).** 5%–30% indicates the proportion of frames replaced by noise.

| Noisy Type | Frame Range | 5% | 10% | 15% | 30% |
|---|---|---|---|---|---|
| drop | 0–30% | 68.2 | 68.3 | 68.2 | 68.5 |
| drop | 30–60% | 68.4 | 68.4 | 68.4 | 68.3 |
| drop | 60–90% | 68.6 | 68.5 | 68.7 | 68.8 |
| modify | 0–30% | 67.2 | 67.1 | 66.6 | 66.4 |
| modify | 30–60% | 66.8 | 65.5 | 64.8 | 63.4 |
| modify | 60–90% | 66.2 | 64.7 | 63.5 | 62.0 |
| Avg $\Delta$ | | -1.22 | -1.92 | -2.49 | -3.16 |

## A.4 ROBUSTNESS ANALYSIS

Our quantification reveals several key robustness characteristics. First, the model demonstrates high tolerance to missing detections (*i.e.*, the *drop* noise type), maintaining stable performance across all noise rates (seen in table 5 and table 6). This indicates that OOTSM can operate effectively even when SGG tools fail to detect certain objects—a common occurrence in challenging visual conditions. Second, we quantify the impact of incorrect detections (*i.e.*, the *modify* noise type), which represent a more severe failure mode. For short-term predictions (R@10), performance degradation scales with noise rate, and temporal analysis reveals that noise in later frames (60–90% of the sequence) has a disproportionate impact: a 30% noise rate in the last segment causes a 4.9% performance drop, compared to only 1.8% when noise is confined to early frames (0–30%). This quantifies the model's reliance on recent observations for immediate predictions.

Most significantly, our noise injection experiments demonstrate remarkable robustness for long-term predictions. Even at an average 25.0% frame error rate (seen in table 7), long-term performance (R@50) degrades by merely 0.68% on average. This showcases OOTSM's ability to maintain prediction accuracy despite substantial input noise, validating its practical applicability in scenarios where perfect SGG detection is unrealistic.

We further evaluate robustness across different SGG quality levels by testing OOTSM with detectors of varying accuracy. The results (seen in table 8) show consistent performance gains regardless of the base SGG tool's accuracy: for a lower-quality detector (DSGDet++), adding OOTSM yields an absolute improvement of 25.9 points in R@50, while even a stronger baseline (STTran) sees a 12.3-point gain. This demonstrates that OOTSM maintains its effectiveness across a wide range of SGG accuracies, from DSGDet's lower baseline performance to STTran's higher baseline. Notably, SceneSayerSDE (which uses STTran internally) achieves comparable short-term performance (60.3% at R@10) but shows limitations in long-term predictions (61.9% at R@50), further validating OOTSM's temporal reasoning capabilities.

Table 6: **Effect of noise type on long-term OOTSM performance (R@50).**

| Noisy Type | Frame Range | 5% | 10% | 15% | 30% |
|---|---|---|---|---|---|
| drop | 0–30% | 73.4 | 73.4 | 73.5 | 73.6 |
| drop | 30–60% | 73.4 | 73.5 | 73.4 | 73.5 |
| drop | 60–90% | 73.6 | 73.7 | 73.8 | 73.9 |
| modify | 0–30% | 73.1 | 73.2 | 73.0 | 73.3 |
| modify | 30–60% | 73.3 | 73.2 | 72.6 | 72.7 |
| modify | 60–90% | 73.4 | 73.0 | 72.9 | 71.6 |
| Avg Δ | | -0.32 | -0.36 | -0.54 | -0.68 |

Table 7: **Average SGG frame error rate after noise injection.**

| | 5% | 10% | 15% | 30% |
|---|---|---|---|---|
| Frame error rate | 5.1% | 9.1% | 13.1% | 25.0% |

## A.5 INPUT ENHANCEMENT STRATEGIES FOR LSGA

To evaluate the potential for improving linguistic scene graph anticipation performance through input modification strategies, we conduct two complementary investigations that examine one-shot prompting with API-based models and natural language caption augmentation independently in Sections A.5.1 and A.5.2.

### A.5.1 ONE-SHOT PROMPTING STRATEGY

We evaluate the performance of several API-based large language models on the pure text-based LSGA task under both zero-shot and one-shot prompting conditions, with results presented in table 9. In the zero-shot setting, models predict future scene graphs based solely on the task description and observed context, while in the one-shot setting, they are additionally provided with a single example illustrating the desired input-output format and task execution. The comparison aims to understand the extent to which these general-purpose models benefit from a minimal demonstration for this specific anticipation task.

The results indicate that incorporating a single example generally enhances the performance of the API-based models. For instance, GPT4o-mini exhibits substantial gains across all metrics when transitioning from zero-shot to one-shot prompting; its R@10 improves markedly from 47.5% to 60.4%, and its mR@10 increases from 31.6% to 41.4%, suggesting the example significantly aids in initial relationship recall and potentially format adherence. Similarly, DeepSeek-V3 demonstrates consistent improvements in both Recall@K and meanRecall@K metrics in the one-shot setting compared to its zero-shot counterpart, with R@20 rising from 57.9% to 63.3% and mR@20 increasing from 46.8% to 50.2%.

GPT4o also benefits from the one-shot example in terms of standard recall, achieving higher R@10, R@20, and R@50 scores (*e.g.*, R@20 improves from 63.9% to 70.3%). Interestingly, however, while its mR@10 improves from 43.3% to 48.1% in the one-shot setting, its mR@20 and mR@50 scores slightly decrease compared to the zero-shot condition (from 52.5% to 48.7% and 52.8% to 48.9%, respectively). This nuanced result for GPT4o suggests that while the one-shot example helps improve overall prediction accuracy (Recall), it might inadvertently shift the model's predictions away from rarer relationship classes, thus slightly lowering the class-balanced meanRecall for longer prediction horizons (K=20, 50). Overall, these findings underscore the utility of in-context examples for adapting large API-based models to specialized tasks like LSGA, although the impact on balanced performance across relationship classes can vary.

While this one-shot analysis provides insights into general-purpose API models' adaptability, our fine-tuned OOTSM model demonstrates robust capabilities under zero-shot conditions. As shown in table 9, OOTSM achieves state-of-the-art zero-shot performance, surpassing even the one-shot performance of competing models on several key metrics. This underscores the effectiveness of our fine-tuning approach for the LSGA task, making separate one-shot evaluation of OOTSM less critical for demonstrating its advantages.

### A.5.2 CAPTION AUGMENTATION STRATEGY

We examine whether appending natural-language captions as environmental context improves linguistic scene graph anticipation without additional fine-tuning. Concretely, for each test video (90% observation; 1,750 videos), we generate a scene description and a human–object interaction description using Qwen2.5-VL-72B and append them to the original task prompt. Unless otherwise stated, decoding follows the main setting (temperature = 0.7, top-$p$ = 0.4), with a maximum context budget of 2048 tokens.

Table 8: **Performance with different SGG tools.**

| Method | R@10 | R@20 | R@50 |
|---|---|---|---|
| DSGDet++ | 38.1 | 39.8 | 39.8 |
| DSGDet + OOTSM | **59.5** | **65.7** | **65.7** |
| SceneSayerSDE (w/ STTran) | 60.3 | 61.9 | 61.9 |
| STTran + OOTSM | **60.6** | **73.2** | **74.2** |

Table 9: **One-shot and zero-shot performance on pure text-based LSGA (w/ GOA).** Each model first predicts future objects before performing relation reasoning. R@K and mR@K ($K = 10, 20, 50$) are reported. **Bold** indicates the best result, and underline indicates the second-best result.

| Mode | Method | R@10 | R@20 | R@50 | mR@10 | mR@20 | mR@50 |
|---|---|---|---|---|---|---|---|
| One-Shot | GPT4o-mini | 60.4 | 63.5 | 63.5 | 41.4 | 49.4 | 49.6 |
| | GPT4o | 66.5 | 70.3 | 70.4 | **48.1** | 48.7 | 48.9 |
| | DeepSeek-V3 | 60.2 | 63.3 | 63.3 | 41.8 | 50.2 | 50.5 |
| Zero Shot | GPT4o-mini | 47.5 | 49.5 | 49.6 | 31.6 | 37.3 | 37.5 |
| | GPT4o | 60.6 | 63.9 | 64.0 | 43.3 | 52.5 | 52.8 |
| | DeepSeek-V3 | 55.1 | 57.9 | 57.9 | 39.7 | 46.8 | 47.1 |
| | OOTSM (Llama-3.2-3B Instruct) | **68.4** | **73.6** | **73.6** | 41.5 | **56.0** | **56.0** |

The quantitative analysis reveals that direct caption concatenation yields only marginal performance improvements. As shown in Table 12, overall recall metrics (R@10/20/50) demonstrate minimal changes within ±0.1%. Notably, while adding detailed scene descriptions achieves modest improvements in short-term performance (mR@10 increases by 3.3%), it fails to enhance long-term prediction results, with OOTSM even performing slightly better on long-horizon metrics. This phenomenon indicates two important insights: (i) temporal reasoning represents an extremely challenging task—although detailed information may provide certain advantages for short-term prediction tasks, it also introduces noise that adversely affects long-term task performance; (ii) structured inputs are beneficial for LLMs performing time-series reasoning. Therefore, even when confronted with multimodal information, how to effectively extract and apply information to temporal reasoning remains a crucial direction for future research.

The computational overhead analysis further highlights the complexity of this problem. Captions were generated using the vLLM framework distributed across 4×NVIDIA H20 80GB GPUs with tensor parallelization. At batch size 1, the captioning process introduces an average wall-clock latency of 11.33 seconds per video, including inter-GPU communication and RPC overhead inherent to the vLLM framework. Given the negligible net improvements demonstrated in Table 12, this represents a substantial computational and latency burden with minimal returns, particularly for long-horizon tasks where R@50 and mR@20/50 show no meaningful improvement. Combined with the above analysis, these findings indicate that for long-horizon LSGA tasks, high-density information does not necessarily translate into effective performance gains, whether from the perspective of prediction accuracy or computational resource utilization efficiency. This result emphasizes that in multimodal scene graph prediction tasks, computational resources should be prioritized for structured reasoning enhancement methods rather than simply increasing multimodal information density.

To illustrate the caption augmentation approach, we provide a representative example corresponding to the scenarios detailed in Section A.8. Each video receives two complementary descriptive components: an environment-level scene description and a human–object interaction description, formatted as follows:

```
1.  Scene Description:
The video is set in a cozy indoor environment, likely a home.  The room features a dining
area with a table covered by a yellow cloth, surrounded by chairs.  There's a decorative
wall with a tree mural and some hanging ornaments.  A small cabinet with various items on
top is visible, along with a small table holding a few objects.  The floor is tiled, and
the overall setting appears lived-in and casual.

2.  Human-Object Interactions:
A person wearing a dark jacket and jeans enters the frame from the right side.  They bend
down near the cabinet, reaching for something on the floor.  The individual then picks up
a long, thin object, which appears to be a broom or similar cleaning tool.  After standing
up, they begin sweeping the floor, moving the broom back and forth across the tiles.  Their
movements are deliberate as they clean the area around the cabinet and under the table.
The person continues sweeping, ensuring the floor is clear of any debris.  Throughout
the sequence, their focus remains on the task at hand, demonstrating a routine cleaning
activity within the home.
```

Table 10: **Object prediction performance.** Evaluation under Temperature=0.7 and Top-p=0.4 averaged over three runs (mean $\pm$ std, in %).

| Partial Acc. | Strict Acc. | Partial Overlap | No Overlap |
|:---:|:---:|:---:|:---:|
| $95.54 \pm 0.18$ | $46.45 \pm 0.00$ | $6.03 \pm 0.36$ | $2.13 \pm 0.00$ |

Table 11: **Relationship prediction performance.** Evaluation under Temperature=0.7 and Top-p=0.4 averaged over three runs (mean ± std, in %).

| Attention Acc. | Spatial Acc. | Contact Acc. | Overall Acc. |
|:---:|:---:|:---:|:---:|
| $69.67 \pm 0.40$ | $74.85 \pm 0.95$ | $71.91 \pm 0.52$ | $72.22 \pm 0.17$ |

## A.6 MODULE PERFORMANCE ANALYSIS

To provide comprehensive insights into the performance characteristics of our two-stage pipeline, we conduct detailed modular analyses that examine both the GOA and OORA modules independently in Sections A.6.1 and A.6.2.

### A.6.1 ANALYSIS OF GOA RESULTS

To quantitatively evaluate object anticipation performance in GOA, we established multiple metrics that capture various dimensions of prediction accuracy. Our evaluation framework incorporates: **Partial Accuracy**, which measures the proportion of frames where predicted object sets contain at least one object from the ground truth; **Strict Accuracy**, which quantifies the proportion of frames with exact matches between predicted and ground truth object sets; **Partial Overlap Rate**, indicating the proportion of frames where predictions partially overlap with ground truth without achieving complete matching or containment; and **No Overlap Rate**, measuring the fraction of frames where predictions fail to identify any ground truth object.

Based on three independent runs (Temperature=0.7, Top-p=0.4; decoding procedures and thresholds selected on the validation set and held fixed on the test set), the model demonstrates consistent object anticipation performance, as detailed in table 10. It achieves substantial coverage (Partial Accuracy of 95.54%) with infrequent complete failures (No Overlap of 2.13%). Exact set matches (Strict Accuracy) are attained at 46.45%, suggesting potential for further boundary calibration. The low Partial Overlap rate (6.03%) indicates that most non-exact outcomes stem from conservative set-boundary decisions rather than ambiguous identity confusions. Reported standard deviations of "$\pm 0.00$" result from rounding to two decimal places (actual std $< 0.005$).

These metrics characterize GOA as recall-oriented and stable, which supports effective transfer of candidate entities to downstream relation reasoning in OORA. Future work may investigate decision calibration techniques, such as adaptive thresholding, validation-calibrated priors, or lightweight post-hoc methods, to improve the precision of set predictions while maintaining coverage levels.

### A.6.2 ANALYSIS OF OORA RESULTS

Table 11 demonstrates that OORA achieves consistent, high-precision relationship decoding across all predicate categories. Spatial relations exhibit the highest accuracy ($74.85\% \pm 0.95$), followed by contact ($71.91\% \pm 0.52$) and attention ($69.67\% \pm 0.40$) relations; the minimal standard deviations (all $< 1\%$) substantiate the reproducibility of performance under the specified decoding parameters. This hierarchical pattern aligns with the empirical observation that spatial configurations typically present less ambiguity than transient gaze directions or subtle contact states. Nevertheless, all three categories converge to a balanced overall accuracy of $72.22\% \pm 0.17$, indicating that our per-object prompting strategy combined with transition regularization enables effective handling of fine-grained temporal dynamics while maintaining robust performance.

When considered alongside the object-level diagnostics presented in table 10, these findings elucidate the substantial end-to-end improvements reported in table 1. The GOA module provides recall-oriented object sets (95.5% partial recall, 2.1% total misses), ensuring comprehensive forwarding of ground-truth entities to OORA. Subsequently, OORA's discriminative architecture and KL-based smoothness loss effectively mitigate the modest over-prediction introduced upstream, enhancing relational precision—particularly for long-tail predicates that predominate the mean-Recall metrics. Consequently, our two-stage pipeline transforms GOA's broad coverage into a significant 4–5% mR@20 improvement observed for Llama-3.2-3B in table 1, while preserving the high Recall@50 crucial for extended-horizon forecasting.

Table 12: **Effect of zero-shot caption prompting (Qwen2.5-(VL)-72B via vLLM).** Captions are appended to the original prompt without fine-tuning. Evaluation on 1,750 test videos (90% observation).

| Method | R@10 | R@20 | R@50 | mR@10 | mR@20 | mR@50 |
|---|---|---|---|---|---|---|
| OOTSM w/o caption | 68.4 | 73.6 | 73.6 | 41.5 | 56.0 | 56.0 |
| OOTSM w/ caption | 68.5 | 73.5 | 73.5 | 44.8 | 55.8 | 56.0 |

## A.7 INFERENCE TIME ANALYSIS

Runtime evaluations were conducted on a single NVIDIA A100-40GB GPU using FP32 precision with batch size = 1 and key-value caching disabled to establish baseline performance metrics. The experimental protocol processed 90% of frames in each video sequence for anticipation of the remaining 10%, with context length constrained to 2k tokens. This configuration enables direct assessment of OOTSM's core linguistic reasoning capabilities for scene graph anticipation without deployment-specific optimizations.

The system processed 1750 samples with mean inference time of 17.44 s per sample, yielding throughput of 3.44 samples/min. Resource utilization measurements recorded peak GPU memory consumption of 39.22 GB (98.1% of available capacity), mean GPU utilization of 76.5%, and computational intensity of 237.1 TFLOPS. These metrics reflect the computational requirements of the unoptimized baseline implementation. Standard deployment optimizations including INT8 quantization, LoRA weight integration, key-value caching, and multi-GPU distribution can substantially improve both latency and throughput characteristics Jacob et al. (2018). For real-time applications, streaming output generation provides additional response time reduction for downstream processing pipelines Xiao et al. (2023).

## A.8 PROMPT AND RESPONSE EXAMPLES

### A.8.1 PROMPT EXAMPLES

To illustrate the linguistic formulation adopted in our pipeline, we provide the complete prompts fed to the large-language model at each stage of inference. GOA formulates object-level forecasting, whereas OORA refines per-object relational dynamics. All text is shown verbatim; line-breaks and escape characters correspond to the exact strings seen by the model.

Below is the GOA' prompt, which instructs the model to predict future object sets:

```
You are an object prediction assistant for scene understanding.  In this task, you are
provided with observed scene information from past frames and a list of future frame
numbers.  Your task is to predict the possible objects for the exact future frames and
answer in a fixed format.

# Example only used in one-shot setting
Example:
Observed:
Frame 42:  object:  medicine attention:  looking_at, spatial:  in_front_of, contact:
holding.
Frame 87:  object:  medicine attention:  looking_at, spatial:  in_front_of, contact:
holding.
object:  cup/glass/bottle attention:  looking_at, spatial:  in_front_of, contact:  holding,
touching.
Frame 111:  object:  medicine attention:  looking_at, spatial:  in_front_of, contact:
holding.
Future frame numbers to predict objects for:  Frame 125, 136
Frame 125:  medicine, cup/glass/bottle
Frame 136:  medicine, cup/glass/bottle

IMPORTANT: Objects may appear or disappear over time.  Consider the following:
1. Objects that were recently visible may still be present even if not mentioned
2. New objects may appear as the scene changes
3. Some objects may disappear from view as time progresses
4. The longer the time gap, the more likely the scene has changed significantly

Available objects:  __background__, person, bag, bed, blanket, book, box, broom, chair,
closet/cabinet, clothes, cup/glass/bottle, dish, door, doorknob, doorway, floor, food,
groceries, laptop, light, medicine, mirror, paper/notebook, phone/camera, picture, pillow,
refrigerator, sandwich, shelf, shoe, sofa/couch, table, television, towel, vacuum, window
Observed:

Frame 16:  object:  table attention:  looking_at, spatial:  in_front_of, contact:
not_contacting.
Frame 47:  object:  table attention:  unsure, spatial:  in_front_of, contact:
```

```
not_contacting.
Frame 77: object: table attention: not_looking_at, spatial: above, contact: touching.
Frame 82..98: object: table attention: not_looking_at, spatial: in_front_of, contact:
touching.
Frame 107: object: table attention: unsure, spatial: in_front_of, contact:
not_contacting.
Frame 114: object: table attention: unsure, spatial: in_front_of, contact: touching.
Frame 129: object: table attention: not_looking_at, spatial: behind, contact:
not_contacting.
Frame 138: object: table attention: unsure, spatial: in_front_of, contact:
not_contacting.
Frame 145: object: table attention: not_looking_at, spatial: behind, contact:
not_contacting.
Frame 168: object: floor attention: looking_at, spatial: beneath,in_front_of, contact:
standing_on.
object: broom attention: looking_at, spatial: in_front_of, contact: not_contacting.
Frame 187: object: floor attention: looking_at, spatial: beneath, contact:
standing_on.
object: broom attention: looking_at, spatial: in_front_of, contact: holding.
Frame 195..205: object: floor attention: looking_at, spatial: beneath,in_front_of,
contact: standing_on.
object: broom attention: looking_at, spatial: in_front_of, contact: holding.
Frame 219: object: floor attention: looking_at, spatial: beneath, contact:
standing_on.
object: broom attention: looking_at, spatial: in_front_of,on_the_side_of, contact:
holding.
Frame 223: object: floor attention: looking_at, spatial: beneath,in_front_of, contact:
standing_on.
object: broom attention: looking_at, spatial: on_the_side_of, contact: holding.
Frame 242: object: floor attention: unsure, spatial: beneath, contact: standing_on.
object: broom attention: looking_at, spatial: in_front_of,on_the_side_of, contact:
holding.
Frame 261: object: floor attention: unsure, spatial: beneath, contact: standing_on.
object: broom attention: looking_at, spatial: in_front_of, contact: holding.
object: doorway attention: not_looking_at, spatial: in, contact: not_contacting.
Frame 262: object: floor attention: looking_at, spatial: beneath, contact:
standing_on.
object: broom attention: looking_at, spatial: in_front_of,on_the_side_of, contact:
holding.
object: doorway attention: not_looking_at, spatial: in, contact: not_contacting.
Frame 268: object: floor attention: looking_at, spatial: beneath, contact:
standing_on.
object: broom attention: not_looking_at, spatial: on_the_side_of,in_front_of, contact:
holding.
object: doorway attention: looking_at, spatial: in, contact: not_contacting.
Frame 275: object: floor attention: looking_at, spatial: beneath, contact:
standing_on.
object: broom attention: looking_at, spatial: in_front_of, contact: holding.
object: doorway attention: not_looking_at, spatial: in, contact: not_contacting.
Frame 285: object: floor attention: looking_at, spatial: above, contact: standing_on.
object: broom attention: looking_at, spatial: in_front_of,on_the_side_of, contact:
holding.
object: doorway attention: looking_at, spatial: in, contact: not_contacting.
Frame 289: object: floor attention: looking_at, spatial: beneath,in_front_of, contact:
standing_on.
object: broom attention: not_looking_at, spatial: in_front_of, contact: holding.
object: doorway attention: not_looking_at, spatial: behind, contact: not_contacting.
Frame 302: object: floor attention: looking_at, spatial: beneath,in_front_of, contact:
standing_on.
object: broom attention: looking_at, spatial: in_front_of,on_the_side_of, contact:
holding.
object: doorway attention: unsure, spatial: in, contact: not_contacting.
Frame 316: object: floor attention: not_looking_at, spatial: in_front_of,beneath,
contact: standing_on.
object: broom attention: looking_at, spatial: in_front_of, contact: holding.
object: doorway attention: looking_at, spatial: in, contact: not_contacting.
Frame 326: object: floor attention: looking_at, spatial: beneath,in_front_of, contact:
standing_on.
object: broom attention: looking_at, spatial: in_front_of, contact: holding.
Frame 389: object: floor attention: unsure, spatial: beneath,in_front_of, contact:
standing_on.
object: broom attention: not_looking_at, spatial: in_front_of, contact: holding.
Frame 413..442: object: floor attention: looking_at, spatial: beneath,in_front_of,
contact: standing_on.
object: broom attention: not_looking_at, spatial: in_front_of, contact: holding.
Frame 452: object: floor attention: looking_at, spatial: beneath, contact:
standing_on.
object: broom attention: not_looking_at, spatial: in_front_of, contact: holding.
Frame 461..480: object: floor attention: looking_at, spatial: beneath,in_front_of,
contact: standing_on.
```

```
object:  broom attention:  looking_at, spatial:  in_front_of, contact:  holding.
Frame 484:  object:  floor attention:  looking_at, spatial:  beneath,in_front_of, contact:
standing_on.
object:  broom attention:  looking_at, spatial:  in_front_of, contact:  not_contacting.

Please output in the following format:
Frame <index>:  <objects>
Each frame should be on a separate line with no additional commentary.

Future frame numbers to predict objects for:  Frame486, Frame 499, Frame 518:
```

GOA predicted two objects—*floor* and *broom*—so we build one prompt per object to forecast its future interactions. Each prompt is fed into OORA independently.

Prompt for [floor]:

```
You are a scene graph anticipation assistant.  In scene graph anticipation, you are given
a series of observed frames containing a specific object.  Your task is to predict how a
person will interact with this object in the future.
Note:
Attention indicates whether the person is looking at the object.
Contact indicates whether the person physically touches or interacts with the object.
Spatial indicates the relative spatial position of the object with respect to the person.

# Example only used in one-shot setting
Example:  Observed segment for object medicine:
Frame 42..207:  object:  medicine attention:  not_looking_at, spatial:  in_front_of,
contact:  holding.
Frame 222:  object:  medicine attention:  not_looking_at, spatial:  in_front_of, contact:
holding.
Future frames:  Frame 226, 236 for object [medicine]:
Frame 226:  medicine attention:  not_looking_at, spatial:  in_front_of, contact:  holding.
Frame 236:  medicine attention:  not_looking_at, spatial:  in_front_of, contact:  holding,
eating.

The possible relationship categories are:
Attention:  looking_at, not_looking_at, unsure
Spatial:  above, beneath, in_front_of, behind, on_the_side_of, in
Contact:  carrying, covered_by, drinking_from, eating, have_it_on_the_back, holding,
leaning_on, lying_on, not_contacting, other_relationship, sitting_on, standing_on, touching,
twisting, wearing, wiping, writing_on

Observed segment for object [floor]:
Frame 168:  object:  floor attention:  looking_at, spatial:  beneath,in_front_of, contact:
standing_on.
Frame 169..187:  object:  floor attention:  looking_at, spatial:  beneath, contact:
standing_on.
Frame 188..205:  object:  floor attention:  looking_at, spatial:  beneath,in_front_of,
contact:  standing_on.
Frame 206..219:  object:  floor attention:  looking_at, spatial:  beneath, contact:
standing_on.
Frame 220..223:  object:  floor attention:  looking_at, spatial:  beneath,in_front_of,
contact:  standing_on.
Frame 224..261:  object:  floor attention:  unsure, spatial:  beneath, contact:
standing_on.
Frame 262..275:  object:  floor attention:  looking_at, spatial:  beneath, contact:
standing_on.
Frame 276..285:  object:  floor attention:  looking_at, spatial:  above, contact:
standing_on.
Frame 286..302:  object:  floor attention:  looking_at, spatial:  beneath,in_front_of,
contact:  standing_on.
Frame 303..316:  object:  floor attention:  not_looking_at, spatial:  beneath,in_front_of,
contact:  standing_on.
Frame 317..326:  object:  floor attention:  looking_at, spatial:  beneath,in_front_of,
contact:  standing_on.
Frame 327..389:  object:  floor attention:  unsure, spatial:  beneath,in_front_of, contact:
standing_on.
Frame 390..442:  object:  floor attention:  looking_at, spatial:  beneath,in_front_of,
contact:  standing_on.
Frame 443..452:  object:  floor attention:  looking_at, spatial:  beneath, contact:
standing_on.
Frame 453..484:  object:  floor attention:  looking_at, spatial:  beneath,in_front_of,
contact:  standing_on.

Please generate the scene graph for object [floor] in each of the following future frames:
486, 499, 518.
Output one scene graph per frame in the following format:
```

```
Frame <index>: object: floor attention: <attention_relationship>, spatial:
<spatial_relationship>, contact: <contact_relationship>
Ensure each frame is on a separate line and no additional commentary is included.

Future frames 486, 499, 518 for object [floor]:
```

Prompt for [broom]:

```
You are a scene graph anticipation assistant.  In scene graph anticipation, you are given
a series of observed frames containing a specific object.  Your task is to predict how a
person will interact with this object in the future.
Note:
Attention indicates whether the person is looking at the object.
Contact indicates whether the person physically touches or interacts with the object.
Spatial indicates the relative spatial position of the object with respect to the person.

# Example only used in one-shot setting
Example:  Observed segment for object medicine:
Frame 42..207:  object:  medicine attention:  not_looking_at, spatial:  in_front_of,
contact:  holding.
Frame 222:  object:  medicine attention:  not_looking_at, spatial:  in_front_of, contact:
holding.
Future frames for object [medicine]:  Frame 226, 236
Frame 226:  medicine attention:  not_looking_at, spatial:  in_front_of, contact:  holding.
Frame 236:  medicine attention:  not_looking_at, spatial:  in_front_of, contact:  holding,
eating.

The possible relationship categories are:
Attention:  looking_at, not_looking_at, unsure
Spatial:  above, beneath, in_front_of, behind, on_the_side_of, in
Contact:  carrying, covered_by, drinking_from, eating, have_it_on_the_back, holding,
leaning_on, lying_on, not_contacting, other_relationship, sitting_on, standing_on, touching,
twisting, wearing, wiping, writing_on

Observed segment for object broom:
Frame 168:  object:  broom attention:  looking_at, spatial:  in_front_of, contact:
not_contacting.
Frame 169..205:  object:  broom attention:  looking_at, spatial:  in_front_of, contact:
holding.
Frame 206..219:  object:  broom attention:  looking_at, spatial:  in_front_of,on_the_side_of,
contact:  holding.
Frame 220..223:  object:  broom attention:  looking_at, spatial:  on_the_side_of, contact:
holding.
Frame 224..242:  object:  broom attention:  looking_at, spatial:  in_front_of,on_the_side_of,
contact:  holding.
Frame 243..261:  object:  broom attention:  looking_at, spatial:  in_front_of, contact:
holding.
Frame 262:  object:  broom attention:  looking_at, spatial:  in_front_of,on_the_side_of,
contact:  holding.
Frame 263..268:  object:  broom attention:  not_looking_at, spatial:
in_front_of,on_the_side_of, contact:  holding.
Frame 269..275:  object:  broom attention:  looking_at, spatial:  in_front_of, contact:
holding.
Frame 276..285:  object:  broom attention:  looking_at, spatial:  in_front_of,on_the_side_of,
contact:  holding.
Frame 286..289:  object:  broom attention:  not_looking_at, spatial:  in_front_of, contact:
holding.
Frame 290..302:  object:  broom attention:  looking_at, spatial:  in_front_of,on_the_side_of,
contact:  holding.
Frame 303..326:  object:  broom attention:  looking_at, spatial:  in_front_of, contact:
holding.
Frame 327..452:  object:  broom attention:  not_looking_at, spatial:  in_front_of, contact:
holding.
Frame 453..480:  object:  broom attention:  looking_at, spatial:  in_front_of, contact:
holding.
Frame 481..484:  object:  broom attention:  looking_at, spatial:  in_front_of, contact:
not_contacting.

Please generate the scene graph for object [broom] in each of the following future frames:
486, 499, 518.
Output one scene graph per frame in the following format:
Frame <index>:  object:  broom attention:  <attention_relationship>, spatial:
<spatial_relationship>, contact:  <contact_relationship>
Ensure each frame is on a separate line and no additional commentary is included.

Future frames 486, 499, 518 for object [broom]:
```

### A.8.2 GENERATION EXAMPLES

After identifying the candidate objects in GOA, we refine each object's interaction trajectory in OORA. The following boxes show GPT-4o-mini's object-level anticipations for both *floor* and *broom*, with separate prompts and outputs for each entity.

GPT-4o-mini GOA object candidates:

```
Frame 486:  floor, broom
Frame 499:  floor, broom
Frame 518:  floor, broom
```

GPT-4o-mini OORA predictions:

```
# Floor predictions
Frame 486:  object:  floor attention:  looking_at, spatial:  beneath,in_front_of, contact:
standing_on
Frame 499:  object:  floor attention:  not_looking_at, spatial:  beneath,in_front_of,
contact:  standing_on
Frame 518:  object:  floor attention:  unsure, spatial:  beneath,in_front_of, contact:
standing_on

# Broom predictions
Frame 486:  object:  broom attention:  not_looking_at, spatial:  in_front_of, contact:
holding.
Frame 499:  object:  broom attention:  looking_at, spatial:  in_front_of, contact:
holding.
Frame 518:  object:  broom attention:  looking_at, spatial:  on_the_side_of, contact:
holding.
```

GPT-4o GOA object candidates:

```
Frame 486:  floor, broom
Frame 499:  floor, broom
Frame 518:  floor, broom
```

GPT-4o OORA predictions:

```
# Floor predictions
Frame 486:  object:  floor attention:  looking_at, spatial:  beneath,in_front_of, contact:
standing_on
Frame 499:  object:  floor attention:  looking_at, spatial:  beneath,in_front_of, contact:
standing_on
Frame 518:  object:  floor attention:  looking_at, spatial:  beneath,in_front_of, contact:
standing_on

# Broom predictions
Frame 486:  object:  broom attention:  looking_at, spatial:  in_front_of, contact:
not_contacting
Frame 499:  object:  broom attention:  looking_at, spatial:  in_front_of, contact:  holding
Frame 518:  object:  broom attention:  not_looking_at, spatial:  in_front_of, contact:
holding
```

DeepSeek-V3 GOA object candidates:

```
Frame 486:  floor, broom
Frame 499:  floor, broom
Frame 518:  floor, broom
```

DeepSeek-V3 OORA predictions:

```
# Floor predictions
Frame 486:  object:  floor attention:  looking_at, spatial:  beneath,in_front_of, contact:
standing_on
Frame 499:  object:  floor attention:  looking_at, spatial:  beneath,in_front_of, contact:
standing_on
Frame 518:  object:  floor attention:  looking_at, spatial:  beneath,in_front_of, contact:
standing_on

# Broom predictions
Frame 486:  object:  broom attention:  looking_at, spatial:  in_front_of, contact:  holding
Frame 499:  object:  broom attention:  looking_at, spatial:  in_front_of, contact:  holding
```

```
Frame 518:  object:  broom attention:  not_looking_at, spatial:  in_front_of, contact:
not_contacting
```

OOTSM GOA object candidates:

```
Frame 486:  floor, broom
Frame 499:  floor, broom
Frame 518:  floor, broom, doorway
```

OOTSM OORA predictions:

```
# Floor predictions
Frame 486:  object:  floor attention:  looking_at, spatial:  beneath,in_front_of, contact:
standing_on.
Frame 499:  object:  floor attention:  looking_at, spatial:  beneath, contact:
standing_on.
Frame 518:  object:  floor attention:  looking_at, spatial:  beneath, contact:
standing_on.

# Broom predictions
Frame 486:  object:  broom attention:  not_looking_at, spatial:  on_the_side_of, contact:
holding.
Frame 499:  object:  broom attention:  not_looking_at, spatial:  on_the_side_of, contact:
holding.
Frame 518:  object:  broom attention:  not_looking_at, spatial:  on_the_side_of, contact:
holding.

# Doorway predictions
Frame 518:  object:  doorway attention:  not_looking_at, spatial:  in, contact:
not_contacting.
```

In fact, the ground-truth for these frames contains only the broom object; its relationships are as follows:

```
Frame 486:  object:  broom attention:  looking_at, spatial:  in_front_of, contact:
not_contacting.
Frame 499:  object:  broom attention:  not_looking_at, spatial:  in_front_of, contact:
holding.
Frame 518:  object:  broom attention:  looking_at, spatial:  in_front_of, contact:
not_contacting.
```

### A.8.3  SUCCESS AND FAILURE CASE STUDIES

Building upon Sections A.8 and A.8.2, this section presents representative success and failure cases spanning both the GOA and OORA stages, accompanied by rigorous academic analysis for each example. To enhance analytical clarity, each case study consolidates consecutive frames containing identical scene graph states and extracts the most informative temporal segments for detailed examination. This collection of cases encompasses the temporal evolution of object sets, gradual trajectories of relationship predicates, and several representative boundary scenarios (including brief disappearances, low-frequency new object entries, and multi-label relationships) to characterize the specific performance of the decoupled architecture in long-horizon prediction contexts.

**Success Cases**   The following examples demonstrate that across extended temporal spans, GOA can generate object set predictions consistent with scene evolution, while OORA captures smooth transitions in relationships such as attention states while maintaining stability in spatial and contact relationships.

GOA Object Prediction — Example S1

```
# Last frames
Frame 586-591:  closet/cabinet, door, doorway
Frame 628-664:  doorway, shoe
Frame 679-802:  doorway

# Predicted frames
Frame 818-830:  doorway
Frame 832-836:  doorway, broom, door
```

In this example, stable structural elements (doorway) are consistently preserved while contextually appropriate re-entry predictions such as broom are generated, demonstrating smooth object set updates that balance temporal consistency with environmental plausibility.

GOA Object Prediction — Example S2

```
# Last frames
Frame 425:  closet/cabinet, clothes
Frame 473:  clothes
Frame 520-523:  clothes, phone/camera
Frame 568:  phone/camera
Frame 572-663:  clothes, phone/camera

# Predicted frames
Frame 720-747:  phone/camera, clothes
Frame 858-970:  phone/camera
```

This case exhibits long-term object persistence combined with gradual decay of secondary objects, conforming to the regularities of temporal evolution while avoiding overfitting to transient fluctuations. The selective persistence demonstrates sophisticated understanding of object importance hierarchies in scene dynamics.

OORA Relationship Prediction — Example S3

```
# Last frames
Frame 661:  object:  floor; attention:  unsure; spatial:  beneath; contact:  standing_on
Frame 662:  object:  floor; attention:  looking_at; spatial:  beneath; contact:
standing_on
Frame 688:  object:  floor; attention:  unsure; spatial:  beneath; contact:  standing_on

# Predicted frames
Frame 691-693:  object:  floor; attention:  unsure; spatial:  beneath; contact:
standing_on
Frame 716:  object:  floor; attention:  not_looking_at; spatial:  beneath; contact:
standing_on
```

This example demonstrates that while spatial and contact relationships remain stable, attention relationships exhibit temporally consistent gradual changes, avoiding unrealistic abrupt transitions. The smooth progression from "unsure" to "not looking at" reflects sophisticated modeling of attention dynamics.

OORA Relationship Prediction — Example S4

```
# Last frames
Frame 864:  object:  table; attention:  unsure; spatial:  in_front_of; contact:
not_contacting
Frame 865:  object:  table; attention:  looking_at; spatial:  in_front_of; contact:
not_contacting
Frame 870-881:  object:  table; attention:  unsure; spatial:  in_front_of; contact:
not_contacting

# Predicted frames
Frame 904:  object:  table; attention:  not_looking_at; spatial:  beneath; contact:
not_contacting
Frame 916:  object:  table; attention:  unsure; spatial:  in_front_of; contact:
not_contacting
```

In this case, spatial relationships exhibit reasonable short-term deviations (beneath) while contact relationships maintain consistency. The coherence across different predicates indicates structured representation of scene geometry and interaction states.

**Failure Cases** The following examples concentrate on boundary scenarios: temporal localization of brief disappearances, low-frequency but plausible new object entries, and multi-label relationship determination, reflecting common yet challenging aspects of long-horizon prediction.

GOA Object Prediction — Example F1 (Inaccurate disappearance timing)

```
# Last frames
Frame 496-499:  book, doorway, paper/notebook, shelf
Frame 527:  book, doorway, shelf
Frame 529:  book, doorway, shelf
Frame 530:  doorway, shelf (book disappears)

# Predicted frames
Frame 533-540:  book, shelf, doorway (incorrect retention)
Frame 584-628:  shelf, doorway (corrected)
Frame 672:  doorway

# Ground truth
Frame 533:  shelf, doorway
Frame 540:  doorway
Frame 672:  doorway
```

This case reveals temporal misalignment in GOA inference, where the timing inaccuracy—treating transient absence at Frame 530 as noise—reflects the inherent challenge of distinguishing momentary transitions in dynamic scenes. This validates the significance of the LSGA task while highlighting opportunities for enhanced temporal sensitivity and change detection mechanisms.

GOA Object Prediction — Example F2 (Missed new object "food")

```
# Last frames
Frame 432:  cup/glass/bottle
Frame 434:  blanket, clothes
Frame 442:  blanket, cup/glass/bottle
Frame 445:  blanket, clothes
Frame 467-578:  clothes

# Predicted frames
Frame 613-816:  clothes (missed food appearance)
Frame 901:  clothes, cup/glass/bottle, phone/camera (incorrect action inference)

# Ground truth
Frame 613-816:  food, clothes
Frame 901:  food, clothes
```

This case illustrates the inherent challenge of state randomness in scene graph anticipation tasks, validating both the complexity and significance of LSGA. Specifically, while dining-related objects (blankets, cups/bottles) provide contextual cues for food appearance, the model incorrectly predicted photography activities at Frame 901. Addressing this randomness requires enhancing physical commonsense reasoning, integrating structured knowledge representations, and incorporating human intention recognition as prior conditions. Although linguistic priors occasionally misalign with plausible but infrequent outcomes, the model's ability to generate contextually coherent predictions remains valuable for downstream applications such as proactive task planning.

OORA Relationship Prediction — Example F3 (Missed multi-label spatial)

```
# Last frames
Frames 529-538:  object:  bag; attention:  not_looking_at; spatial:  on_the_side_of;
contact:  holding
Frame 541:  object:  bag; attention:  not_looking_at; spatial:  in_front_of, on_the_side_of;
contact:  holding
Frame 584:  object:  bag; attention:  not_looking_at; spatial:  on_the_side_of; contact:
holding

# Predicted frames
Frames 631-658:  object:  bag; attention:  not_looking_at; spatial:  on_the_side_of;
contact:  holding

# Ground truth
Frames 631-658:  object:  bag; attention:  not_looking_at; spatial:  in_front_of,
on_the_side_of; contact:  holding
```

This case demonstrates the model's selective relationship modeling capability—accurately predicting attention and contact relationships while struggling with multi-label spatial scenarios. This challenge underscores the significance of LSGA in capturing nuanced relational states and suggests that multi-label optimization strategies could further enhance the framework's predictive capabilities. The collapse of complex spatial relationships into simpler single-label predictions reflects a fundamental limitation in handling combinatorial relationship complexity that warrants future investigation.

These case studies reveal that the decoupled GOA-to-OORA pipeline successfully leverages structured representations to model complex temporal dynamics while highlighting specific areas for improvement. The success cases validate our design hypothesis that separating object and relationship prediction allows each component to specialize in its respective temporal characteristics. However, the failure cases concentrate on boundary conditions that expose current limitations in change detection, rare event prediction, and multi-label relationship modeling, providing clear directions for future enhancements to the framework.

## A.9 LLM Usage Statement

LLMs were used in this work primarily to aid and polish the writing of this paper. Specifically, LLMs were employed for grammar correction, sentence restructuring, and improving the clarity of technical descriptions while preserving the original research contributions and findings. All technical content, experimental results, analysis, and conclusions were generated by the authors. The authors take full responsibility for the accuracy and validity of all content presented in this paper.

