# OpenReview forum: "OOTSM: A Decoupled Linguistic Framework for Effective Scene Graph Anticipation"
_ICLR.cc/2026/Conference — ICLR 2026 Conference Withdrawn Submission_

### Official Review · Reviewer_Cs29 · 2025-10-19

**Soundness:** 2
**Presentation:** 2
**Contribution:** 3
**Rating:** 4
**Confidence:** 4

**Summary:**

This paper introduces OOTSM, a two-stage framework for Linguistic Scene Graph Anticipation (LSGA), a novel variant of the Scene Graph Anticipation (SGA) problem. The key idea is to decouple visual perception from semantic temporal reasoning, which is reformulated as a text-based forecasting task handled by a fine-tuned LLM. The proposed method consists of two modules: Global Object Anticipation (GOA), which predicts future object sets; and Object-Oriented Relationship Anticipation (OORA), which forecasts object-specific relationships with temporal consistency.
Experiments on the Action Genome dataset show that fine-tuned compact open-source LLMs (e.g., Llama 3B) outperform strong zero-shot models such as GPT-4o in long-horizon prediction and that integrating OOTSM with visual detectors improves standard video-based SGA metrics.

**Strengths:**

1. The problem formulation of Linguistic Scene Graph Anticipation (LSGA) is novel, interesting and solid. LSGA is a clean and modular reformulation of SGA that disentangles vision from reasoning. This conceptual clarity is valuable and could inspire new research directions bridging symbolic reasoning and video understanding.

2. The two-stage OOTSM design is well-motivated, addressing both context window limits in small LLMs and temporal inconsistency in relation prediction.

3. GOA provides a novel solution to the continuous-object limitation in SGA.

4. The ablation studies are thorough and insightful. The authors isolate the effects of cosine-weighted losses and transition regularization, and provide sensitivity analyses over observation window size and weighting parameter β.

**Weaknesses:**

1. The novelty in modeling architecture is limited.

1.1 While the problem formulation is novel, the core technical contributions (weighted CE loss, KL-based transition regularizer) are relatively straightforward extensions rather than fundamentally new learning paradigms.

1.2 As a measure of novelty, OOTSM's performance on video based SGA is not significantly better than SceneSayer. It is not surprising that OOTSM outperforms SceneSayer on mR@20/50 because OOTSM introduces LLMs which naturally address the long-tail distribution issue in SGA. It is concerning why OOTSM cannot outperform SceneSayer on R@10, especially for longer anticipation windows (F=0.3, 0.5), despite the usage of LLMs.

1.3 This also raises concerns on the necessity of using LLMs. Which part of the performance gain is actually contributed by the LLMs? What are OOTSM's training / inference efficiency and computational resource cost compared to SceneSayer?

2. The experiments are not enough sufficient.

2.1 The evaluation on video based SGA is not complete. The authors only evaluates on the easiest setting GAGS, where ground truth objects and bounding boxes are available. OOTSM's performance under more challenging settings PGAGS (only GT bounding box available) and AGS (no GT available) is not evaluated, limiting the impact of this research.

2.2 The performance of GOA should be explicitly evaluated since it is one of the core contributions of this paper. How well does GOA detect appearing / disappearing objects?

2.3 The results under the No Constraint SGA setting are not provided.

3. The writing clarity of Section 3.3 and 3.4 could be improved by explaining Figure 3 and Figure 4 in more detail in the main text. While the figures themselves are informative, the captions are not, and there is no further explanation of the workflow of GOA and OORA in the main text.

4. Qualitative analysis is very limited, and the paper could benefit from showcasing success / failure in temporal reasoning.

**Questions:**

See Weaknesses 1.2, 1.3 and 2.2.

---

> ### Author Response · Authors · 2025-11-15
> **Response to Questions (W1.2, W1.3, and W2.2)**
>
> **I. W1.2** **Performance analysis**
>
> We need to clarify the performance data interpretation. SceneSayer employs raw video features with higher information density for scene dynamics, excelling in short-term forecasting. However, the high-density dynamic information also introduces more noise, potentially leading to performance degradation in long-term prediction to some extend. **Our method's original intention is to explore whether decoupling the SGA task into the linguistic modality and leveraging LLM's semantic reasoning capabilities improves prediction performance.** The data in Table 2 fully validates this design.
>
> For R@10, although slight declines occur at F=0.3/0.5, improvements are achieved at F=0.7/0.9. More importantly, **on the mR@10 metric that considers long-tail distribution, our method improves across all windows: 1.3, 1.1, 2.7, and 3.4 at F=0.3/0.5/0.7/0.9,respectively.** For medium and long-term metrics, **the performance gains brought by LLMs are even more significant, reaching up to 21.6.**
>
> These results demonstrate LLM's clear advantages in addressing object category imbalance and medium-to-long-term prediction, representing our paper's core contribution. **Except for two short-term metrics, we show superiority across all evaluation dimensions.** The results validate the linguistic paradigm while revealing a promising direction: enhancing short-term prediction by leveraging video features' high information density while maintaining linguistic reasoning advantages. This represents an important future direction.
>
> **II. W1.3** **LLM necessity and computational efficiency**
>
> Regarding the quantification of LLM contributions, we analyze from two perspectives.
>
> **First, the performance gains of OOTSM over traditional VLM methods directly reflect the value of LLMs.** Table 2 shows the comparison with SceneSayer, a Transformer-based video method. At F=0.9, OOTSM improves mR@50 from 29.8 to 51.7 (+21.9) and consistently outperforms SceneSayer across all long-term prediction metrics.
>
> **Second, the comparison between OOTSM and closed-source LLM APIs in Table 1 and Table 9 (Appendix A.5.1) validates the effectiveness of our fine-tuning strategy.** Using the 3B-parameter Llama model with specialized GOA and OORA module designs, cosine-weighted CE loss, and transition loss training strategy, OOTSM (R@20/50=73.6/73.6) not only surpasses larger-scale closed-source models in zero-shot settings (e.g., GPT4o: R@20=63.9/64.0) but also outperforms one-shot configurations (e.g., GPT4o: R@20/50=70.3/70.4).
>
> Regarding computational efficiency, we conducted comprehensive evaluation on a single NVIDIA A100-SXM4-40GB GPU, with results shown in Table as follows:
>
> | **Metric** | **Value** |
> | --- | --- |
> | Processed samples | 1,750 |
> | Average processing time | 17.44 seconds/sample |
> | Peak GPU memory usage | 39.22 GB / 40 GB (98.1%) |
> | Average computational intensity | 237.1 TFLOPS |
>
> The 3B-parameter model enables single-GPU training and inference without distributed systems or API calls. Our objective is exploring the linguistic paradigm's potential, which current results validate. Future engineering optimizations such as INT8 quantization, LoRA merging, and streaming output can further improve efficiency[1][2].
>
> **III. W2.2 GOA module performance evaluation**
>
> Thank you for the suggestion. We provide evaluation GOA's performance in predicting appearing/disappearing objects.
>
> We designed two metrics for GOA evaluation: **New Object Appearance Accuracy (New%)** measures prediction of newly appearing objects, and **Object Disappearance Accuracy (Dis%)** measures prediction of object disappearance. Both are calculated based on video-level object set unions relative to the baseline object set in the last frame of the observation window.
>
> Evaluation results on sequences containing object dynamics:
>
> **Table 13. Performance evaluation of GOA on object appearance and disappearance.**
>
> | **Method** | **New (%)** | Dis (**%)** | **Avg (%)** |
> | --- | --- | --- | --- |
> | GOA w/ cosine-weighted CE loss | 32.1 | 85.4 | 58.8 |
> | GOA w/o cosine-weighted CE loss | 25.7 | 84.3 | 55.0 |
>
> The results demonstrate two key findings. **First, cosine-weighted CE loss significantly improves object dynamics prediction**, achieving 6.4% improvement in new object detection and 1.1% in disappearance detection (average 3.8%), consistent with Table 3's ablation study. **Second, GOA achieves 58.8% average accuracy, demonstrating its capability to address the non-continuous objects challenge.** New object detection (32.1%) is lower due to the difficulty of predicting entirely new objects, while disappearance detection reaches 85.4% since disappearing objects come from the known observation window.
>
> [1] Jacob B, et al. Quantization and training of neural networks for efficient integer-arithmetic-only inference[C]. CVPR, 2018.
>
> [2] Xiao G, et al. Efficient streaming language models with attention sinks[J]. arXiv, 2023.

---

> > ### Author Response · Authors · 2025-11-15
> > **Response to Remaining Weaknesses**
> >
> > **I. W1.1 Problem formulation and framework design.**
> >
> > Thank you for your feedback. We emphasize the core value of our contributions:
> >
> > **First**, we introduce **LSGA for the first time**, representing a paradigm shift that decoupled the SGA task into a linguistic framework. Based on this novel problem definition, we provide an effective solution that achieves R@20=73.6/mR@20=56.0 on text-based LSGA and successfully transfers to video-based SGA (mR@50 improves from 29.8 to 51.7 at F=0.9, +21.9).
> >
> > **Second**, our method focuses on **systematic framework design**. Targeting the non-continuous objects challenge present in the dataset, we propose a two-stage solution:
> >
> > 1. **GOA module**: The first stage handles object dynamics by predicting changes in the future object set. Cosine-weighted CE loss down-weights distant tokens to stabilize training, achieving a 4.2 improvement in mR@20 (from 51.8 to 56.0).
> > 2. **OORA module**: The second stage performs relationship prediction on the predicted object set. Transition loss enforces smoothness in relationship distributions across adjacent time steps, enhancing temporal consistency (R@20/50 from 72.6 to 73.3) and improving long-tail prediction (mR@20/50 from 53.9 to 54.9).
> >
> > **The systematic ablation in Table 3 validates the necessity and complementarity of each component**, with the combined version achieving optimal performance. Our value lies in providing a systematic and effective framework design for our proposed LSGA task.
> >
> > **II. W2.1 & W2.3 Evaluation settings**
> >
> > Thank you for your attention to the evaluation settings. We would like to clarify the rationale behind our metric choices.
> >
> > **First, regarding the choice of GAGS for video-based SGA evaluation.** PGAGS and AGS primarily evaluate **object detection capabilities in SGG tasks**, which is why they are more challenging. However, this is not the focus of our research. **Our core contribution is decoupling the LSGA task from SGA and validating the linguistic paradigm for scene graph temporal reasoning.** We have conducted comprehensive evaluations on the LSGA task. Table 1 presents text-based LSGA evaluation, which is the core validation of linguistic reasoning. Additionally, we provide multi-dimensional analyses including video-based SGA transfer validation (Table 2), ablation studies (Table 3), different observation ratios (Figure 5a), and qualitative analysis (Appendix A.8.3). We welcome other works to compare on the LSGA benchmark.
> >
> > **Second, regarding the choice of the No Constraint metric.** With Constraint selects top-K predictions per object pair based on GT count, providing the most direct assessment of model prediction performance. In contrast, No Constraint performs global ranking using obj_scores × rel_scores to select top-100 triplets.
> >
> > Our LLM outputs discrete labels converted to near **one-hot distributions**, resulting in nearly identical With Constraint and No Constraint results. **This is because LLMs are designed to produce categorical outputs rather than graded confidence distributions required for global ranking.** No Constraint evaluation assumes models can provide comparable confidence scores for all triplets, which fundamentally mismatches the discrete decision nature of LLMs. **Therefore, comparison under No Constraint is unfair for the LLM-based OOTSM model.**
> >
> > **III. W3 & W4 Writing clarity and qualitative analysis**
> >
> > Thank you for the valuable suggestions.
> >
> > **Regarding W3 (Figure 3 and Figure 4 clarity).** We have enhanced the captions for Figures 3 and 4 in the revised version to provide comprehensive workflow descriptions. The updated captions now detail the processing flow of observed scene graphs and the collaborative training process of loss components. **The revised PDF has been uploaded.**
> >
> > **Regarding W4 (Qualitative analysis).** We have actually provided detailed qualitative analysis in Appendix A.8.3. Success cases demonstrate OOTSM's capabilities in object dynamics prediction and relationship transition modeling, validating effective temporal reasoning. Failure cases analyze model limitations on rare relationship categories and extremely long-horizon predictions. These cases thoroughly showcase both the advantages and challenges of the linguistic paradigm in temporal reasoning. For more detailed discussion, please refer to our manuscript.

---

### Official Review · Reviewer_mKJT · 2025-10-31

**Soundness:** 3
**Presentation:** 2
**Contribution:** 2
**Rating:** 4
**Confidence:** 3

**Summary:**

The paper reframes Scene Graph Anticipation (SGA) as a linguistic forecasting problem (LSGA) and proposes OOTSM, a two-stage, text-only anticipation framework. The framework includes GOA, a global object-set forecaster trained with cosine-weighted CE over future tokens to down-weight distant horizons; and OORA, an object-oriented relation forecaster with a temporal transition regularizer, a symmetrized KL over transition histograms, and an auxiliary BCE head. The authors construct an LSGA benchmark by converting Action Genome to temporally ordered SG sequences, and they report that small open LLMs outperform zero-shot API models on text-only LSGA, and that plugging OOTSM behind a frozen SGG tool yields strong video-based SGA results, especially for long horizons.

**Strengths:**

This two-stage architecture leverages GOA to overcome context limitations and employs OORA to model per-object multi-label relationships with a principled transition prior, which is technically motivated.

Noise-injection studies distinguish between drop noise and modify noise, revealing that short-term predictions are sensitive to later-frame errors. In contrast, long-term performance remains remarkably stable (only a 0.68% drop at 25% frame errors), highlighting the true source of the model’s signal.

**Weaknesses:**

All core claims are tied to Action Genome. While AG is a standard, results could overfit the dataset priors, limiting generality claims.

The paper compares fine-tuned 1.5–3B models to zero-shot closed-model APIs, which is not an apples-to-apples comparison. The author claims that compact LLMs surpass zero-shot
APIs is plausible, but methodologically conflated with tuning.

GOA weighting and OORA regularization give limited gains. The ablation shows +0.7 on R@20 and +2.1 on mR@20 from the two losses combined, which is useful but small, without deeper analysis across $\beta, \lambda$.

**Questions:**

It seems that in Table 1, the R@50 has the same performance as R@20, and w/ GOA is not noticeably better than w/o GOA. The author should explain why.

Figure 5(b) and Section 4.5 shows that including $\beta$ in the GOA loss has a negligible effect on overall performance. This finding is corroborated by the marginal improvement between w/o GOA and w/ GOA in Table 1.

---

> ### Author Response · Authors · 2025-11-15
>
> **I. W1 Dataset coverage and generalizability.**
>
> Thank you for this concern. We emphasize two key points regarding dataset selection:
>
> **First, Action Genome is the established benchmark for SGA.** All prior SGA methods (including STTran, DSGDet, SceneSayerODE/SDE) evaluate exclusively on this dataset for fair comparison. Action Genome remains the only real-world video dataset with dense frame-level human-object interaction annotations[1][2], making it ideal for temporal scene graph anticipation. Importantly, our quantitative analysis (Appendix A.3.1) reveals that under the standard 90%-10% protocol, only 61% of sequences maintain object consistency, while 39% exhibit object dynamics (14% new objects, 25% disappearances), demonstrating the dataset's research value.
>
> **Second, our robustness analysis demonstrates that the model learns genuine temporal dynamics rather than dataset-specific patterns.** Systematic noise injection experiments (Section A.4, Tables 5-8) show only −0.68% R@50 degradation under 25% frame corruption. **Notably, the model exhibits strong robustness to modify noise in observed frames**, indicating that LSGA relies on temporal causal relationships in scene graphs rather than superficial features. Furthermore, temporal reasoning validation (Appendix A.3.2) and success case analysis (Appendix A.8.3) confirm that the model captures true causal dependencies, not dataset artifacts.
>
> **II. W2 Fair comparison between fine-tuned small Models and zero-shot APIs.**
>
> Thank you for this concern. Our goal is not to claim small models universally outperform closed-source APIs, but to validate a training strategy: **whether task-specific fine-tuning is more effective than general-purpose zero-shot for structured, temporally-constrained LSGA tasks**. Due to computational constraints, we fine-tune only ≤3B open-source LLMs while using larger closed-source models as API baselines.
>
> To address concerns about model capacity differences, we provide one-shot results (Appendix A.5.1): closed-source APIs show significant improvement over their zero-shot versions, yet still underperform our fine-tuned models on most metrics, especially medium-to-long-term horizons. Specifically, GPT-4o (one-shot) achieves R@50=70.4, mR@50=48.9, while OOTSM (Llama-3.2-3B, **fine-tuned**) reaches R@50=73.6, mR@50=56.0. This demonstrates that **the task-specialized fine-tuning strategy itself is effective**, rather than merely a model size effect.
>
> **III. W3 & Q2 Limited gains from GOA weighting and OORA regularization.**
>
> Thank you for your detailed review. We clarify the effectiveness of these design components:
>
> 1. **Regarding the effectiveness of combined losses:**
>
>    Table 1 clearly demonstrates the significant contribution of the GOA module: w/ GOA achieves a **4.2 improvement in mR@20** compared to w/o GOA (from 51.8 to 56.0), which is crucial for long-tail relationship prediction. The ablation study in Table 3 further reveals the complementarity of the two losses. Cosine-weighted CE loss primarily improves long-tail prediction (mR@20 from 53.9 to 54.9) by down-weighting distant-future tokens to reduce noisy gradients. Transition loss enhances temporal consistency (R@20 from 72.6 to 73.3). The success cases in Appendix A.8.3 further demonstrate the model's capability to accurately predict object dynamics.
>
>
>    **It is important to note that when the overall baseline is high and Recall approaches saturation, absolute improvements become compressed.** Furthermore, the boundary case failure cases shown in Appendix A.8.3 reveal the inherent challenges present in the dataset. A detailed summary can be found in our response to Reviewer kZYv.
>
> 2. **Regarding the β parameter and GOA module:**
>
>    Figure 5(b) and Section 4.5 show that performance remains stable across the 0.3-0.7 range, with key metrics varying by less than 0.5. **This indicates that the cosine weighting mechanism is insensitive to the β value**, and researchers can select any value within this range to achieve comparable performance.
>
> **IV. Q1 R@K metric saturation and GOA effectiveness.**
>
> Thank you for this observation. The R@20 and R@50 performance similarity results from the Recall@K ceiling effect. In the Action Genome dataset, each future frame contains limited GT relationships (typically <20). When K exceeds the GT count, Recall@K saturates, which is an inherent metric property rather than method failure.
>
> **Regarding GOA effectiveness, mR@20 reveals the true contribution: w/ GOA achieves 4.2 improvement over w/o GOA (from 51.8 to 56.0).** This demonstrates GOA's effectiveness in long-tail relationship prediction and object dynamics modeling. For detailed analysis, please refer to our response to W3&Q2.
>
> [1]Krishna R, et al. Visual genome: Connecting language and vision using crowdsourced dense image annotations[J]. ICJV, 2017.
>
> [2]Shang X, et al. Annotating objects and relations in user-generated videos[C]. ICMR. 2019.

---

### Official Review · Reviewer_kZYv · 2025-11-01

**Soundness:** 2
**Presentation:** 3
**Contribution:** 2
**Rating:** 2
**Confidence:** 5

**Summary:**

This paper introduces a novel task termed Linguistic Scene Graph Anticipation (LSGA) and proposes an Object-Oriented Two-Staged Method (OOTSM) to address it. The core idea is to decouple the Scene Graph Anticipation (SGA) problem: a visual scene graph detection model first converts a video clip into a sequence of historical scene graphs, and then a purely text-based model, operating on this sequence, predicts future scene graphs.

**Strengths:**

1. Comprehensive Evaluation:The experimental design is thorough. The paper provides convincing evidence on two fronts: first, by rigorously evaluating the LSGA task itself against powerful baselines, and second, by demonstrating that advancements in LSGA directly translate to superior performance on the established video SGA task. The substantial improvement in long-horizon prediction (e.g., +21.9% mR@50) is a strong result that highlights the practical value of this approach.

**Weaknesses:**

1. Task Delineation Requires Emphasis:While LSGA is novel, the paper does not sufficiently articulate the fundamental distinctions between LSGA and existing tasks like Dynamic Scene Graph Generation (SGG) or Video SGGin the main text (particularly in the Introduction and Related Work sections). Existing video SGG tasks primarily focus on describingthe present or past by detecting and linking objects across frames. In contrast, LSGA is fundamentally about anticipationor forecastingfuture states. This shift from "what is/was" to "what will be" is crucial and introduces unique challenges (e.g., heavy reliance on commonsense and causal reasoning) and applications. This distinction should be a central point of the paper's narrative.

2. Dependence on Front-end Performance:The performance of the LSGA module is inherently upper-bounded by the quality of the initial scene graph detection. Errors from the front-end model will inevitably propagate and could be amplified in the forecasting stage. The paper should discuss this limitation more explicitly.

3. Benchmark Details:More detailed information about the constructed LSGA benchmark is needed. Key statistics (e.g., dataset size, train/val/test splits, temporal horizons for prediction, class distributions) and the precise construction methodology are critical for reproducibility and future benchmarking.

**Questions:**

1. As highlighted in the weaknesses, the core differentiator of LSGA is the "anticipation of future states" versus the "description of current/past states" in Video SGG.Could you please elaborate on this distinction more explicitly in the paper? Specifically, what are the unique advantages and challenges of formulating this forecasting problem in the linguistic modality? This is essential for readers to fully grasp the contribution.

2. How does the performance of OOTSM degrade with errors in the input scene graph sequence from the front-end detector? Is there any analysis or inherent robustness in your method to such noisy inputs?

---

> ### Author Response · Authors · 2025-11-14
>
> **I. W1 & Q1 Introduction to the LSGA task and analysis of its advantages and challenges.**
>
> Thank you for your comments. We clarify that Scene Graph Anticipation (SGA) is a **well-established task in prior literature**, defined as predicting future scene graphs from observed videos. LSGA builds upon this task by exploring whether linguistic reasoning via LLMs can enhance SGA performance. This hierarchy is consistently emphasized throughout our paper:
>
> 1. **Abstract**:
>
> > Scene Graph Anticipation (SGA) involves predicting future scene graphs from video clips... We therefore propose Linguistic Scene Graph Anticipation (LSGA) as an independent forecasting task in the language domain
> >
> 2. **Introduction**:
>
> > Given a sequence of observed video frames, SGA aims to predict future scene graphs ... Building on this decoupled strategy... we propose a novel task called LSGA
> >
> 3. **Related Work**:
>
> > SGA extends SGG to temporal prediction over video streams... Our approach uniquely employs LLMs as the exclusive reasoning engine for long-term SGA
> >
>
> **Regarding your question about the advantages and challenges of formulating SGA in the linguistic modality**, we emphasize that **our entire experimental framework (Section 4) and Appendices A.3-A.8 systematically address this core concern**. To facilitate your understanding of our contribution, we provide a concise summary below, with comprehensive details available in the full manuscript.
>
> 1. **Advantages**: Three key benefits demonstrated through experiments
>
>     a. **Enhanced medium-to-long-horizon prediction (Section 4.2, Table 2, F=0.9)**: LSGA achieves **+21.9% mR50** improvement at F=0.9 compared to SceneSayerSDE, with gains concentrated at R20 (+11.3) and R50 (+12.3) metrics, validating LLMs' capacity to leverage commonsense priors for extended temporal reasoning beyond immediate visual evidence
>
>     b. **Robustness to observation uncertainty (Section A.4, Tables 5-8)**: LSGA maintains stable performance with only **0.68% R@50 degradation** under 25% frame error rates (Table 6, 30% noise injection; Table 7 frame error mapping), demonstrating resilience when visual evidence is ambiguous or incomplete
>
>     c. **Superior long-tail prediction (Section 4.2, Table 2, F=0.9)**: The substantial **+18.3 mR@20** improvement at F=0.9 indicates effective handling of rare relationship classes through linguistic knowledge embedded in pretrained LLMs
>
> 2. **Challenges**: Limitations revealed through failure case analysis exploring model performance boundaries **(Appendix A.8.3)**:
>
>     a. The inherently challenging task of distinguishing transient disappearances from continuations is challenged (Example F1)
>
>     b. Anticipating low-frequency yet plausible object emergence in contextually rich scenarios remains difficult (Example F2), where reasonable appearances are occasionally overlooked despite strong cues.
>
>     c. Predicting multi-label spatial relationships exposes combinatorial complexity (Example F3), where compound relations are occasionally collapsed despite accurate attention and contact predictions.
>
>     These challenges represent core difficulties inherent to LSGA, and our boundary case analysis demonstrates to the community the task's research value and potential for future investigation.
>
> **II. W2 & Q2 Front-end error propagation and robustness.**
>
> Thank you for this question. As detailed in our response to **W1 & Q1**, we have systematically addressed noise robustness through comprehensive experiments (Section A.4, Tables 5-8), demonstrating that long-term performance remains remarkably stable with only **−0.68% R@50 degradation** even at 25% frame error rates. Additionally, **Reviewer mKJT has acknowledged this contribution in their strengths, recognizing the model's robust signal under noisy conditions.**
>
> Furthermore, Section 3.5 evaluates performance under both GT scene graphs and detector outputs, while Section A.1.2 details our threshold-aware margin multi-label loss that stabilizes SGG quality. These systematic analyses quantitatively demonstrate OOTSM's robustness to front-end errors and methodologically address the degradation concern.
>
> **III. W3 Benchmark details and reproducibility.**
>
> Thank you for your interest. Section 4 provides comprehensive benchmark specifications: our LSGA benchmark is automatically converted from Action Genome annotations, comprising ~11.4K videos, 35 object classes, and 25 relation classes. We strictly follow the official train-test split without re-sampling, ensuring reproducibility and fair comparison with existing SGA methods. Preprocessing details (e.g., temporal graph merging) are documented in Section 4 and Appendix A.8. We have open-sourced our complete codebase, including all data processing pipelines and implementation details.
>
> We hope these responses have addressed your questions. If you still have any concerns, we are happy to provide additional clarification.

---

### Note · Authors · 2025-12-03

**Comment:**

We thank the reviewers for their time and feedback. After careful consideration, we have decided to withdraw this submission from ICLR 2026.

We appreciate the reviewers' perspectives. We note that several core aspects of our work, including the systematic formulation of LSGA as a controlled benchmark for temporal reasoning, the documented limitations of object continuity assumptions (affecting 39% of videos), the 21.9% improvement in long-horizon mR@50, and the practical robustness demonstrated under realistic detection noise, appear to have received limited emphasis in the evaluation, though these were central to our intended contributions.

We acknowledge that clearer positioning and presentation could better communicate these contributions, and we will continue to refine our work accordingly.

**Withdrawal Confirmation:**

I have read and agree with the venue's withdrawal policy on behalf of myself and my co-authors.